# Extensive 5'-surveillance guards against non-canonical NAD-caps of nuclear mRNAs in yeast

Yaqing Zhang [1], David Kuster [1], Tobias Schmidt [2], Daniel Kirrmaier[3,4], Gabriele Nübel[1], David Ibberson[5], Vladimir Benes [6], Hans Hombauer [2,3], Michael Knop [3,4] & Andres Jäschke [1✉]

The ubiquitous redox coenzyme nicotinamide adenine dinucleotide (NAD) acts as a non-canonical cap structure on prokaryotic and eukaryotic ribonucleic acids. Here we find that in budding yeast, NAD-RNAs are abundant (>1400 species), short (<170 nt), and mostly correspond to mRNA 5'-ends. The modification percentage of transcripts is low (<5%). NAD incorporation occurs mainly during transcription initiation by RNA polymerase II, which uses distinct promoters with a YAAG core motif for this purpose. Most NAD-RNAs are 3'-truncated. At least three decapping enzymes, Rai1, Dxo1, and Npy1, guard against NAD-RNA at different cellular locations, targeting overlapping transcript populations. NAD-mRNAs are not translatable in vitro. Our work indicates that in budding yeast, most of the NAD incorporation into RNA seems to be disadvantageous to the cell, which has evolved a diverse surveillance machinery to prematurely terminate, decap and reject NAD-RNAs.

[1] Institute of Pharmacy and Molecular Biotechnology (IPMB), Heidelberg University, 69120 Heidelberg, Germany. [2] DNA Repair Mechanisms and Cancer, German Cancer Research Center (DKFZ), 69120 Heidelberg, Germany. [3] Zentrum für Molekulare Biologie der Universität Heidelberg (ZMBH), DKFZ-ZMBH Alliance, Heidelberg University, 69120 Heidelberg, Germany. [4] Cell Morphogenesis and Signal Transduction, German Cancer Research Center (DKFZ), DKFZ-ZMBH Alliance, 69120 Heidelberg, Germany. [5] Deep Sequencing Core Facility, CellNetworks, Heidelberg University, 69120 Heidelberg, Germany. [6] Genomics Core Facility, European Molecular Biology Laboratory (EMBL), 69117 Heidelberg, Germany. ✉email: jaeschke@uni-hd.de

n eukaryotes, the 5′-terminus of messenger RNAs is protected by a m[7]-guanosine (m[7]G) cap[1], which modulates pre-mRNA splicing, polyadenylation, nuclear exit, and translation initiation[2]. The cap is hydrolyzed by various decapping enzymes[3], thereby triggering RNA degradation. Recently, another type of 5′-cap structure was discovered, both in prokaryotes[4–6] and eukaryotes[7–10], which is derived from the ubiquitous redox coenzyme NAD. Although in human and plant cells a diverse landscape of NAD-RNAs was found, only 37 RNA species and low abundance were reported in budding yeast[8], questioning the biological significance of NAD capping in this organism. As the protocol used in this work excluded the small RNA fraction, which had been particularly rich in NAD-RNAs in prokaryotes[4], we address here the whole landscape of NAD transcripts in yeast using the original NAD captureSeq protocol[11]. We find that NAD-RNAs are ubiquitous (1400 in wild type (WT), several thousands in mutants), most of them being short species (<170 nt). Only very few RNAs are detected with lengths over 250 nt. Ab initio incorporation by RNA polymerase (RNAP) II is found to be the predominant mechanism for NAD incorporation, and for about half of the transcripts, RNAP II uses transcription start sites (TSSs) different from those utilized for m[7]G-capped RNAs of the same gene. A YAAG core promoter motif is found to correlate with efficient transcriptional NAD incorporation. On average, NAD-RNAs are shorter than non-NAD-RNAs. By deleting the (putative) NAD-RNA decapping enzymes Npy1, Dxo1, and Rai1, we identified overlapping populations of RNAs likely decapped by these enzymes. Sequence analysis of total RNA from each mutant strain supports a hierarchical order of NAD-RNA processing, in agreement with the subcellular locations of the enzymes. NAD-mRNAs that escape decapping do not support translation by cytosolic ribosomes in vitro. We propose that (at least for the nuclear transcripts studied here) NAD incorporation into yeast RNA is largely accidental, due to competition of NAD and ATP in transcription initiation. We speculate that the NAD modification is in most cases undesirable to the cell, which first disfavors the synthesis of full-length NAD-RNAs, then decaps them rapidly using a multi-tiered machinery localized in different compartments and—even if they reach full length and escape decapping—ultimately rejects them from ribosomes.

## Results

**Short NAD-RNAs are abundant in yeast.** To comprehensively address NAD-RNAs in budding yeast, we isolated total RNA from yeast strain BY4742 and applied the original NAD captureSeq protocol[4], in which the enzyme adenosine diphosphate-ribosylcyclase (ADPRC) tags NAD-RNAs at the NAD moiety, followed by click-chemistry biotinylation and selective isolation by streptavidin binding. After adapter ligation, reverse transcription (RT), and PCR amplification, amplicons with sizes between 150 and 300 bp were selected by gel electrophoresis; thus, this library represented mostly RNA species with sizes between 20 and 170 nt present in the original sample. Enrichment was determined by quantitative comparison with a minus ADPRC negative control. In this unfragmented library, 1460 RNAs were found to be enriched, with changes reaching up to 1200-fold (Fig. 1a). Sixty-nine percent of the genome-mapped reads corresponded to mRNA 5′-ends (Fig. 1b), whereas only little enrichment was observed for mRNA fragments starting further downstream (for details and validation, see Supplementary Methods and Supplementary Fig. 1a–d). Small nucleolar RNAs (snoRNAs) and ribosomal RNA (rRNA) fragments comprised 9.6% and 7.1% of the reads, but represented only 2 and 1 different RNA species, respectively. Thirteen percent corresponded to RNA fragments too small for unique genome mapping (12–17 nt,

Fig. 1b), many of which showed homology to enriched members of the mRNA 5′-end group (Supplementary Fig. 1e). To probe the existence of full-length NAD-capped mRNAs, two additional datasets were generated from total RNA that was random-sheared prior to NAD captureSeq using different size selection windows (small fragmented: 20–170 nt; large fragmented: 170–350 nt). Consistently, these fragmented libraries revealed much lower numbers of enriched species and enrichment values (small fragmented: 145 RNAs, maximum fold change (FC) < 7 (Fig. 1c) and large fragmented: 200 RNAs, maximum FC < 9 (Supplementary Fig. 1f)), mainly due to increased background in the minus ADPRC controls. Extensive overlap was detected between the two fragmented libraries and the unfragmented one (83.8% and 76.0%, respectively; Fig. 1e). Five out of the 12 genes explicitly reported in the previous study[8] (Supplementary Fig. 1h) overlap with the enriched species in our unfragmented library (COX2, LSM6, ERG2, UBC7, and YJR112W-A) and two with the fragmented library (LSM6 and UBC7; black dots in Fig. 1a, c).

To test whether highly expressed transcripts are generally more likely to be enriched in NAD captureSeq, we compared the enrichment levels observed in the unfragmented NAD captureSeq library with the transcript abundance determined by transcriptome sequencing. This analysis revealed no correlation (Supplementary Fig. 1i).

As NAD captureSeq enriches the 5′-ends of NAD-RNAs and sequences in 5′–3′ direction, the overall lengths of transcripts larger than the Illumina read length cannot be reliably inferred from the sequencing reads (Fig. 1d). We therefore carried out the first steps of the NAD captureSeq protocol (until the enriched RNAs were bound to streptavidin) and carried out RT quantitative PCR (RT-qPCR) on two RNA species that were enriched in the fragmented and unfragmented libraries (TDH3 and SED1) using four different primer pairs each. These data revealed roughly equal abundance from the 5′-end through ~300 nt, whereas their 3′-untranslated regions (UTRs) were reduced in abundance by several orders of magnitude (Fig. 2a, green bars). The preferential enrichment of smaller RNAs in NAD captureSeq was also confirmed by Bioanalyzer size analysis of the DNA amplicons (Supplementary Fig. 1j).

For two transcripts (TDH3 and POR1 mRNA), the 5′-NAD modification was directly identified and quantified by mass spectrometry (MS) after pulldown (Fig. 2b and Supplementary Fig. 1i), confirming the chemical identity of the NAD modification. Thus, NAD-RNAs are abundant, short, and mostly correspond to mRNA 5′-ends in budding yeast.

**Nudix pyrophosphohydrolase Npy1 processes NAD-RNA.** In *Escherichia coli*, the Nudix hydrolase NudC acts as an efficient decapping enzyme for NAD-RNA[4,12,13]. The yeast homolog Npy1 is known to hydrolyze the pyrophosphate bond in NAD to yield nicotinamide mononucleotide (NMN) and adenosine monophosphate (AMP)[14], and was recently suggested as an NAD-RNA decapping enzyme[13]. The only support for this claim was, however, its in vitro processing of a synthetic NAD-RNA 12mer into a product that migrated on HPLC like a 12mer-5′-monophosphate RNA (p-RNA) and the inactivity of an active-site mutant to produce this product[13]. To characterize the in vitro activity of Npy1, we purified the protein from *E. coli* and analyzed its reaction kinetics with an in vitro-transcribed NAD-RNA (a 98 nt 5′-fragment of TDH3 RNA) on acryloylaminophenyl boronic acid (APB) gels, which separate NAD-RNA from p-RNA[15]. Purified Npy1 decapped NAD-RNA without inducing nucleolytic degradation and had no effect on m[7]G-RNA in vitro (Fig. 3a). Furthermore, efficient decapping of NAD-RNA required Mn[2+] ions (Supplementary Fig. 2a–c). A Npy1 mutant

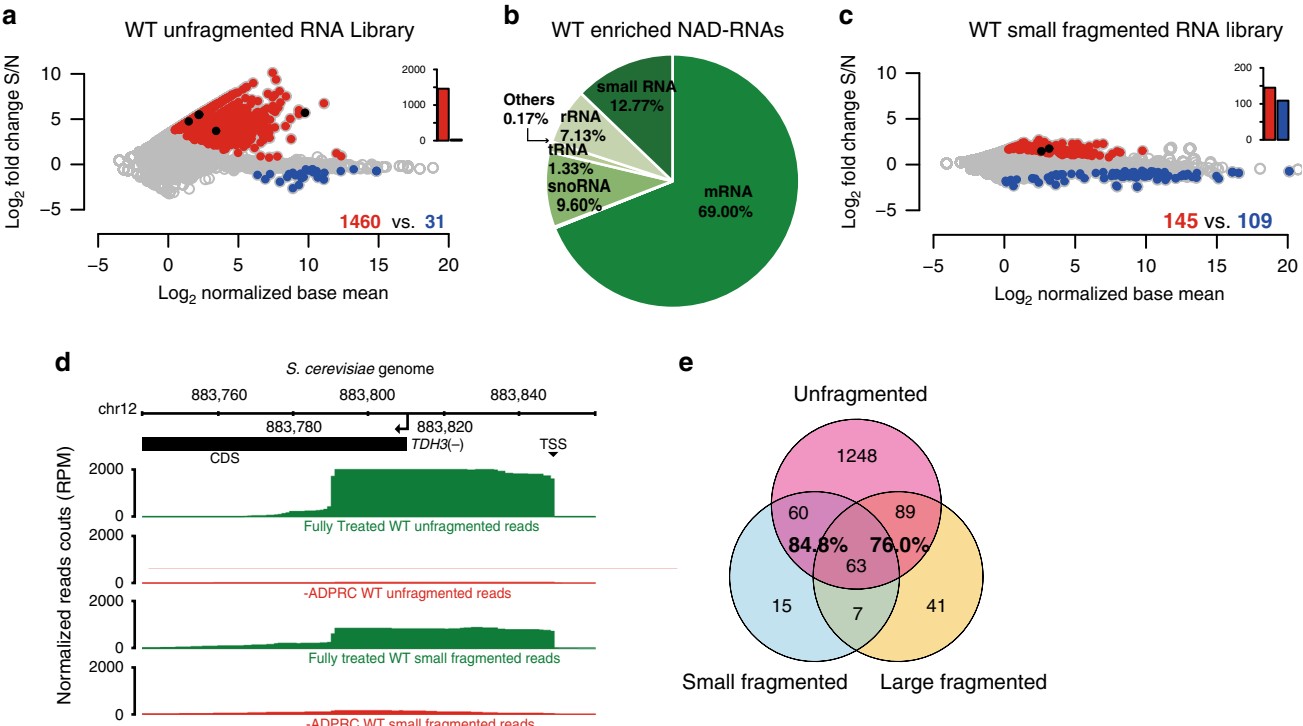

**Fig. 1 NAD-RNAs are abundant in the yeast *S. cerevisiae*. a** Enriched NAD-RNAs as determined by unfragmented NAD captureSeq on the wild-type (WT) strain. The log$_2$ fold change between fully treated sample (S) and minus ADPRC negative control (N) is plotted vs. the log$_2$ normalized base mean. Different transcripts (7620) were analyzed and are represented as a dot. Red dots represent enriched NAD transcripts (fold change (FC) > 1.414, normalized base mean (NBM) > 1, $p < 0.05$); blue dots represent negatively enriched transcripts (FC < 0.707, NBM > 1, $p < 0.05$); black dots denote transcripts previously reported as NAD-capped[8]. The bar chart represents the number of red and blue dots using the same color. Performed in biologically independent replicates, $n = 3$. **b** Reads of enriched NAD-RNA from the WT unfragmented NAD captureSeq library, mapped to genome-annotated RNAs. Chart contains reads classified as genome-annotated RNAs (reads ≥ 18 nt, same statistics parameters as in **a**), plus the self-categorized small RNAs (12 nt ≤ reads < 18 nt, fold change S/N > 1.414). Other categories included the classes pseudogene (0.093%), ncRNA (0.057%), and long-terminal repeat RNA (0.017%). The pie chart represents the number of reads assigned to each RNA class as percentage of total reads. **c** Enriched NAD-RNAs as determined by small fragmented NAD captureSeq on WT strain. Random-sheared total RNA as input and sized selected cDNA (insert < 172 bp) was utilized for library building. Statistics parameters are as in **a**. Performed in biologically independent replicates, $n = 3$. **d** Aligned sequencing reads of the *TDH3* gene are visualized in the integrated genome browser (IGB). The read count was normalized as reads per million genome-mapped reads (RPM). The arrow indicates the transcription direction and the coding sequence is displayed as black bar. The chromosome number and genome locus are denoted at the top. "Fully treated" represents the library with ADPRC treatment, whereas −ADPRC stands for the library obtained by omitting this treatment. **e** Venn diagram displaying the intersection of enriched NAD-RNA species among the unfragmented, small fragmented, and large fragmented WT NAD captureSeq libraries. The percent values are defined as number of shared species, divided by the total number of species in the fragmented library. Source data are provided as a Source Data file.

in which a catalytic glutamate was replaced (E276Q) showed no decapping activity (Supplementary Fig. 2d). In addition to NAD-RNA, Npy1 also hydrolyzed NAD into NMN and AMP in a Mn$^{2+}$-dependent manner (Supplementary Fig. 2e), whereas the E267Q mutant was inactive (Supplementary Fig. 2f). Thus, recombinant Npy1 decaps NAD-RNA in vitro.

To address whether Npy1 also functions on NAD-RNA in vivo, we investigated a yeast strain lacking Npy1. In agreement with the yeast SGA database[16], the absence of Npy1 caused no severe phenotypical changes under a variety of growth conditions (Supplementary Fig. 2g). Although gene expression analysis by transcriptome sequencing indicated changes in abundance for almost 50% of all detected transcripts (Supplementary Fig. 2h), mass spectrometric whole proteome analysis detected only very few proteins with significant (more than twofold) changes in expression, in comparison to the WT strain (Fig. 3b). Deletion of *NPY1* slightly increased the total cellular concentration of NAD (by ~10%, Fig. 3c). When we applied NAD captureSeq to RNA purified from the *npy1Δ* strain, twice as many uniquely mapped RNAs (3028, unfragmented library) were NAD-capped (relative to WT), which were almost half

of all detected RNA species (Fig. 3d). Consistent with the WT, NAD-RNAs from the *npy1Δ* strain were mostly short transcripts, as only 242 and 226 NAD-RNA species were enriched in the small and large fragmented RNA libraries, respectively (Supplementary Fig. 2i, j). Compared to the WT, a similar proportion of the reads allocated to mRNA 5′-ends in the unfragmented library (59.1%) but three times more on very small RNAs (40.3%), suggesting that Npy1 is involved in the decapping of small NAD-RNAs. rRNAs and snoRNAs disappeared almost completely (Fig. 3e).

Contrary to our expectations (but as also observed in *Bacillus subtilis*[5]), removal of Npy1 reduced the total amount of NAD attached to RNA by ~60% (Fig. 4a and Supplementary Fig. 2k). We assessed the change in the apparent modification ratio (percentage of an RNA species that carries NAD) transcriptome-wide by integrating transcriptome and NAD captureSeq data, using the enrichment values in NAD captureSeq as proxy (Fig. 4b). This analysis indicated that upon *NPY1* gene deletion, the NAD-modification ratio was reduced for 1013 species, whereas it increased for 164. There was no correlation between expression level (change) and modification ratio (change).

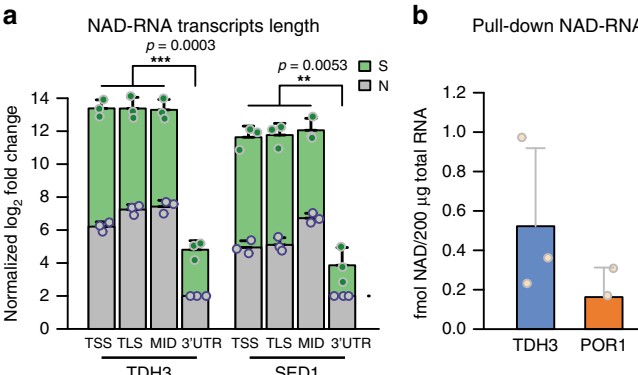

**Fig. 2 Characterization of NAD-RNAs from WT yeast. a** qRT-PCR of NAD-RNAs targeting different regions of the gene: the 5′-end including the transcription start site (TSS), the region around the translation start site (TLS), a middle (MID) region around the first in-frame ATG after the TLS, and the 3′-UTR. Green bar heights represent relative transcript numbers in the sample group (+ADPRC), whereas gray bar heights represent the transcripts numbers from the same species in the negative control group (−ADPRC). Log$_2$ fold change of TLS, TSS, and MID was normalized to the 3′-UTR of $N$ (set as 2). Dots represent the measured values. Error bars represent mean + SD. Performed in biologically independent replicates, $n = 3$. $p$-values are denoted by asterisks: *$p < 0.05$; **$p < 0.01$; ***$p < 0.001$ (the minimum value among TSS, TLS, MID vs. the value from the 3′-UTR, Student's $t$-test, one-sided). **b** Pulled-down NAD-RNA quantified by LC-MS. NAD-RNAs were estimated by measuring the NAD content, which was determined based on the signal intensity of nicotinamide riboside (NR) (fmol) from NudC-treated pulldown RNA subtracted by NR signal intensities, obtained from RNA not treated with NudC. Performed in biologically independent replicates, $n = 3$. Statistics parameters are as in Fig. 2a. Source data are provided as a Source Data file.

Plotting the modification ratio of $npy1\Delta$ mutant vs. WT confirmed that the global reduction of NAD modification is not caused by few strongly reduced species that override the effects of many weakly increased ones. The slope <1 (0.61) of this plot confirms that, on average, the modification in the $npy1\Delta$ mutant is lower than in the WT (Supplementary Fig. 2l). To independently support the decreased modification ratios in the npy1Δ mutant derived from NAD captureSeq, we quantified 18 different RNAs by quantitative RT-PCR (qRT-PCR) in the unfragmented cDNA libraries of sample (S) and negative control (N) for WT and mutant strain. After normalization of the cp values to the same amount of input RNA in WT and $npy1\Delta$, and background subtraction, NAD-modification ratios below 5%, in most cases between 1% and 3% were determined (Fig. 4c). Those 16 genes with decreased NAD modification showed indeed reduced PCR amplification, whereas those two with increased modification (LSM7 and RSM10) PCR-amplified stronger (Fig. 4c). To test whether this increased modification ratio of the latter two transcripts might indicate that they are particularly good substrates for Npy1, being severely depleted in RNA preparation from cells containing this enzyme, we analyzed their decapping in vitro. Indeed, LSM7 and RSM10 were hydrolyzed much faster than TDH3 (Fig. 4d, e). Collectively, these data are consistent with a role of Npy1 in processing NAD caps in vivo.

**Npy1, Rai1, and Dxo1 influence the NAD-RNA landscape.** The non-Nudix enzymes Rai1 and Dxo1 were previously reported to decap NAD-RNA in vitro and in vivo by a mechanism different from Npy1, namely by removal of the entire NAD moiety en bloc[7,17]. To compare the influence of all three enzymes on the global NAD-modification landscape of RNAs in vivo, we created

all possible combinations of $rai1\Delta$, $dxo1\Delta$, and $npy1\Delta$ deletion mutants. Phenotypically, the removal of Rai1 (from the WT and from mutant strains) had the strongest negative effect on growth in normal medium and in the presence of increasing concentrations of ethanol (Fig. 5a and Supplementary Fig. 3a). On the transcriptome level, we detected in all mutant strains ~1000 upregulated and ~1000 downregulated RNA species (at least fourfold, relative to WT), together corresponding to ~30% of all mRNAs (Supplementary Fig. 3b). There was over 60% overlap in regulated genes between the three different single-knockout mutants, whereas ~600 genes were selectively regulated by only one decapping enzyme (Supplementary Fig. 3c). A systematic analysis of the effect of the deletion of one particular enzyme in WT and mutant strains revealed high agreement within one group (e.g., all strains carrying a deletion of the $NPY1$ gene), and the strongest global effect on RNA expression was noticed for removal of $RAI1$ (Fig. 5b). Analysis of total RNA isolated from the knockout mutants by NAD captureSeq revealed enrichment of more than half of all detected RNA species (3765 in $dxo1\Delta$; 3810 in $rai1\Delta$), indicative of their modification with NAD. No significant further increase was observed in the double- and triple- deletion strains (Fig. 5c). In the triple-knockout $dxo1\Delta$ $rai1\Delta$ $npy1\Delta$, only mRNA fragments (63%) and small RNAs (35%) were detected by NAD captureSeq (Fig. 5d). Unlike the WT, the top 250 enriched NAD-RNA species of all mutants functionally clustered (by Gene Ontology terms) as rRNA metabolic process and translation (Supplementary Fig. 3e). Thus, our analysis of the $rai1\Delta$-, $dxo1\Delta$-, and $npy1\Delta$-knockout strains may be consistent with a role of the affected gene products in processing NAD capping, but by no means demonstrative evidence.

**NAD-RNAs have distinct TSSs.** The above analysis suggested that the landscape of NAD-RNA transcripts is shaped by (at least) four enzymes as follows: RNAP II, Rai1, Dxo1, and Npy1. Using the deletion mutants, we first analyzed transcriptional preferences. Although the sequencing read profiles of some RNAs revealed homogenous 5′-ends (indicative of a defined TSS), others showed irregular patterns suggesting pervasive transcription or multiple TSSs (see Fig. 1d and Supplementary Fig. 4a for examples). From the NAD captureSeq data, we selected all significantly enriched RNAs starting with an A which had homogenous 5′-ends ("sharp A" selection). When we compared our experimentally determined 5′-ends of these NAD-RNAs with published next-generation sequencing (NGS)-derived and 5′-rapid amplification of cDNA end-validated TSSs for canonical (i.e., non-NAD-) RNAs[18], for nearly half of all species the 5′-ends differed (Fig. 6a, b). For the WT strain, 98 RNAs were observed in which the 5′-transcript leader (TL) sequences were longer than in the database, whereas 63 species got shorter, in some cases by more than 100 nt (Fig. 6c). This TL length change was not only observed in the WT strain (in both unfragmented and fragmented libraries, Fig. 6c and Supplementary Fig. 4b, c), but also in all mutants, including the $dxo1\Delta rai1\Delta$ $npy1\Delta$ triple mutant (Fig. 6b and Supplementary Fig. 4d), suggesting that RNAP II might select a different TSS for initiating transcription with NAD instead of ATP, compared to the canonical TSS[19,20]. The changed TL length upon NAD incorporation could be corroborated by qRT-PCR with primers targeting either our NAD captureSeq-observed TSS or the canonical ones from the database, comparing the RNAs enriched in NAD captureSeq with a non-enriched total RNA preparation (Fig. 6d). Thus, NAD-RNAs tend to have longer TL sequences than non-NAD-RNAs, indicative of their synthesis starting at a more distal TSS.

**A YAAG promoter motif supports NAD incorporation by RNAP II.** We supposed that analysis of the $dxo1\Delta rai1\Delta npy1\Delta$

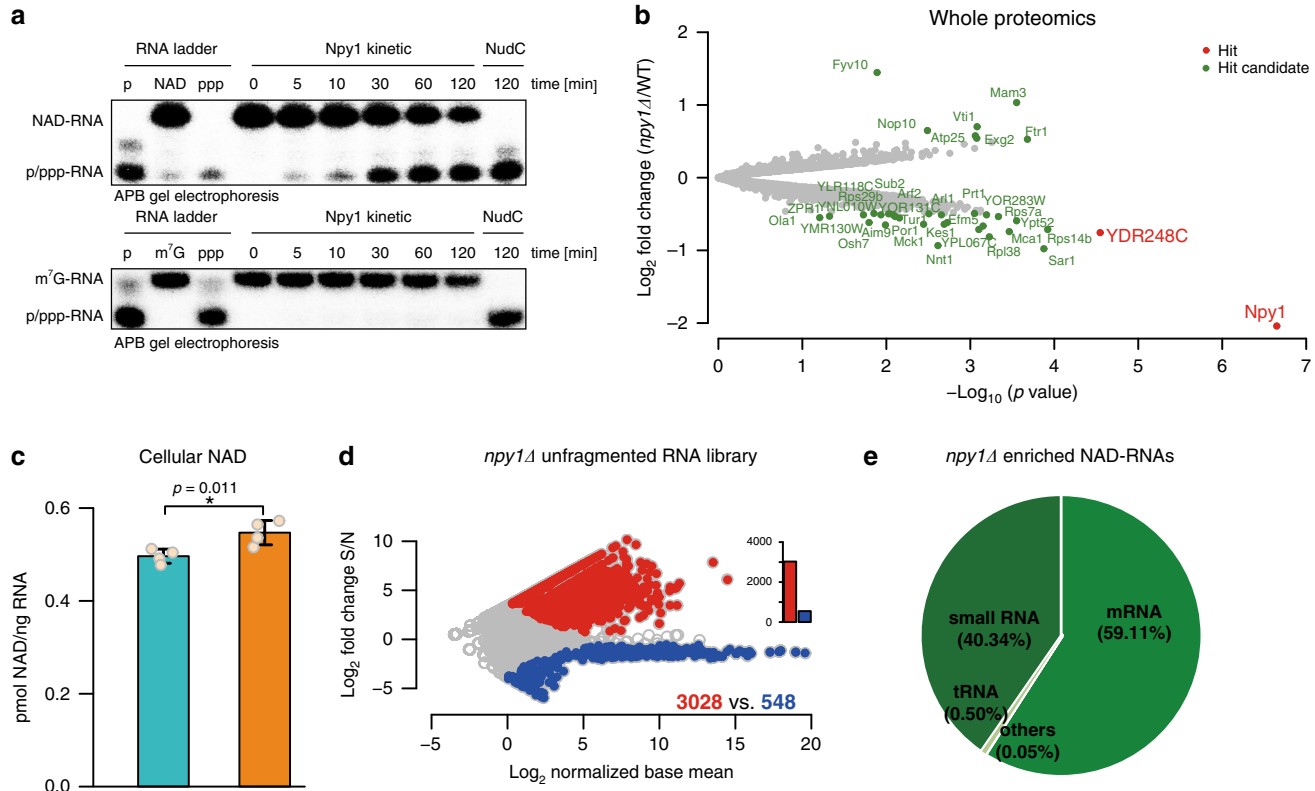

**Fig. 3 Npy1 affects NAD-RNA in vitro and in vivo. a** *S. cerevisiae* Npy1 in vitro enzyme kinetics of decapping of 5′-NAD- and 5′-m$^7$G modified RNAs was analyzed in the presence of Mn$^{2+}$. A α-$^{32}$P-body-labeled 5′-fragment of TDH3 RNA (98 nt) and a corresponding NAD-RNA control were treated with NudC in the presence of 2 mM Mg$^{2+}$ and 1 mM Mn$^{2+}$, and resolved by APB gel electrophoresis. Three independent experiments were performed, $n = 3$. **b** Whole proteome analysis of cell lysates (WT strain vs. *npy1Δ*). Red dots indicate enriched hits (fold change >1.5 and FDR < 0.05), whereas green dots are hit candidates (fold change > 1.4, 0.05 ≤ FDR < 0.2) and gray dots are proteins without significant change. **c** Quantification of cellular NAD content in WT (blue bar) and *npy1Δ* (orange bar) yeast strains, determined using an enzyme cycling assay. The amount of NAD was normalized to the amount of total RNA determined in cell lysates, obtained from each sample. Error bars represent the mean ± SD. Performed in biologically independent replicates, $n = 4$. *p*-values are denoted by asterisks: *$p < 0.05$ (Student's *t*-test, one-sided). **d** Enriched NAD-RNAs in the unfragmented NAD captureSeq library of the Npy1 depletion (*npy1Δ*) strain. Performed in biologically independent replicates, $n = 3$. **e** Reads of enriched NAD-RNA in the *npy1Δ* unfragmented NAD captureSeq library, mapped to genomically annotated RNAs. All parameters as in Fig. 1b. Source data are provided as a Source Data file.

triple-knockout mutant strain would reveal the least biased information about the factors that govern transcriptional NAD incorporation by RNAP II. We mapped nucleotides −10 to +10, relative to the RNA 5′-end inferred from the NAD captureSeq reads, for the 25 most enriched "sharp A" NAD-RNAs (log$_2$FC > 8, false discovery rate (FDR) < 0.00002) and for appropriate control groups (i.e., RNA species not enriched in NAD capture-Seq). In the enriched fraction, we observed a highly conserved motif YAAG (with the first A being the 5′-terminal nucleotide of the transcript, i.e., the site where the NAD is incorporated), followed by an A-rich stretch of lower significance, whereas in the non-enriched fraction no preferences were found (Fig. 7a). The motif was not observed when for the same top 25 candidate RNAs the published canonical TSSs[18] were mapped (Fig. 7b). When for those 25 genes all TSSs listed in the yeast TSS data-base[21] (top 5 abundant TSSs per gene) were analyzed, only a YA motif[22] was identified (Supplementary Fig. 5a). However, when only the TSS (from this database) closest to our observed one was utilized, the YAAG motif appeared prominently (Supplementary Fig. 5b). This analysis implies that our identification procedure revealed real TSSs and further supports the assumption that transcriptional NAD incorporation is the predominant biosynthetic pathway to furnish NAD-RNAs. The YAAG motif was also observed (although less prominently) in the top 100 and 200 enriched RNAs, and its prominence decreased with decreasing

NAD captureSeq enrichment values (Supplementary Fig. 5c, d). It could also be detected in WT and all mutant strains, whereby generally the significance decreased with increasing number of decapping enzymes present (Supplementary Fig. 5e–k). The motif was not observed when the NAD captureSeq-enriched snoRNAs or transfer RNAs (tRNAs) were mapped (Supplementary Fig. 5l, m), suggesting that these candidates may have a different bio-genesis. To exclude the possibility that the motif reflects a bias introduced by the enzymes applied in NAD captureSeq (ADPRC, reverse transcriptase, terminal deoxynucleotidyl transferase, two ligases), we mapped the top 25 enriched sequences from our previously published *E. coli*, *B. subtilis*, and *Staphylococcus aureus* NAD captureSeq datasets by the same procedure, finding neither YAAG nor an A-rich tail (Supplementary Fig. 5n–p). Further analysis revealed that this motif constitutes a fraction of known "good" RNAP II core promoter sequences, having all conserved features[23–25], namely: (1) being A/T-rich between positions −30 and +10, (2) a switch from T-rich to A-rich in the coding strand around position −8, (3) a pyrimidine at position −1, and (4) an A at position +1. In addition, two specific features distinguish good NAD-incorporating promoters, namely a slightly increased probability for an A at position +2 and a strongly conserved G at +3.

To test whether this motif actually modulates NAD incorpora-tion by RNAP II in vivo, we deleted gene *TDH3*, a highly enriched

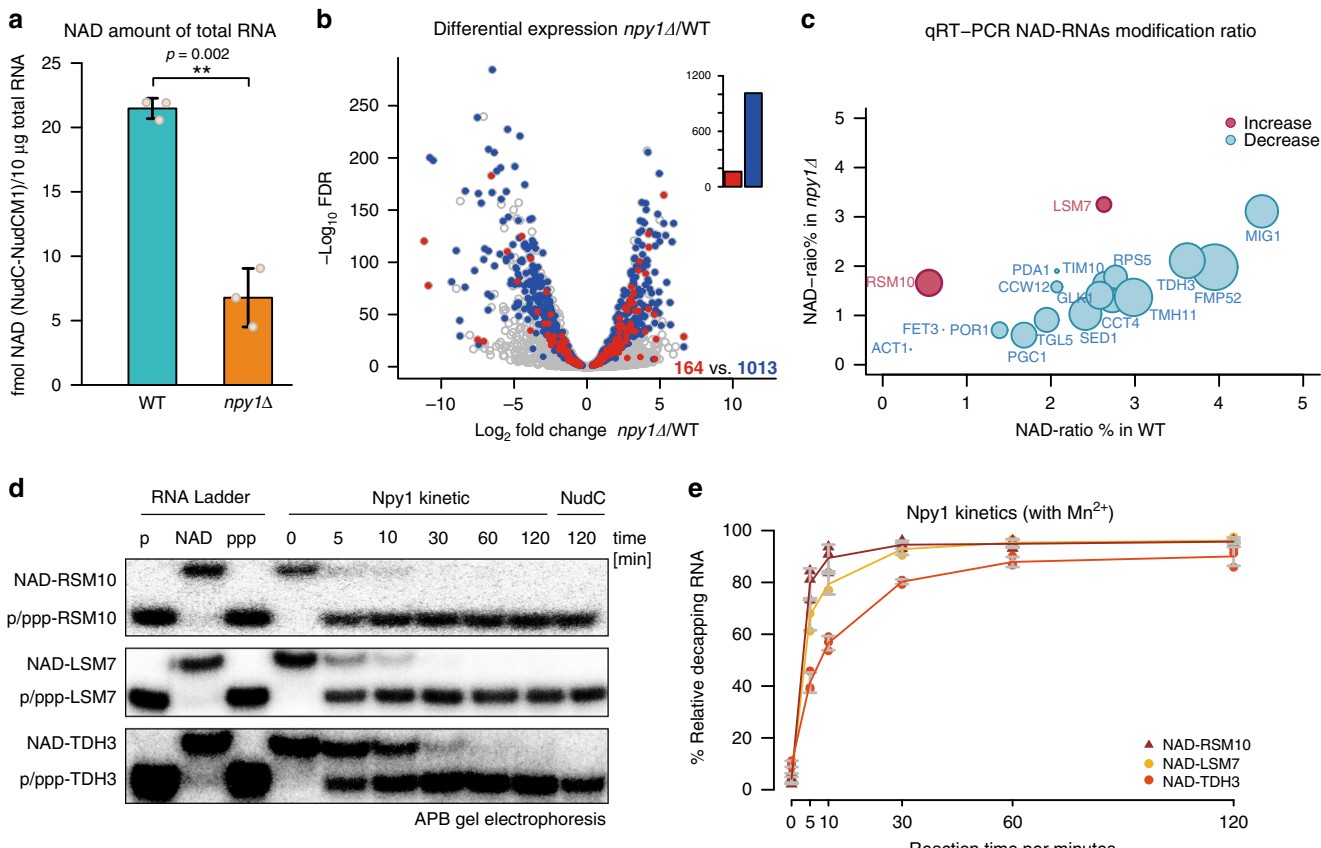

**Fig. 4 RNAs are affected differently by Npy1 deletion. a** Quantification of NAD-RNA from total RNA by LC–MS. Integrated NAD content, which was determined via the NR signal intensity (fmol) from washed and NudC-treated total RNA (10 μg), while subtracting the NR signal intensity obtained from RNA treated with a NudC mutant. Dots represent individual biological triplicate measurements. Error bars represent mean ± SD. Performed in biologically independent replicates, $n = 3$. $p$-values are denoted by asterisks: **$p < 0.01$ (Student's $t$-test, one-sided). **b** Volcano plot of transcripts comparing WT with $npy1\Delta$ samples by RNA expression level and NAD-modification ratio. The $\log_2$ fold change of transcript abundance from transcriptome sequencing ($npy1\Delta$ vs. WT) is plotted versus the $\log_{10}$ false discovery rate (FDR). Different transcripts (7620) were analyzed and represented as dots. Red dots represent RNAs for which the NAD ratio increased upon $NPY1$ gene deletion according to NAD captureSeq (NAD ratio: $npy1\Delta >$ WT $> 0$, transcriptome normalized base mean >100, $p < 0.05$, FDR < 0.1), and blue dots represent RNAs for which the NAD ratio decreased (NAD ratio: WT $> npy1\Delta > 0$, transcriptome normalized base mean >100, $p < 0.05$, FDR < 0.1). **c** Bubble plot of the relative NAD-modification ratio of 18 RNA species, as determined by qRT-PCR. Blue bubbles represent RNAs for which the NAD-modification ratio decreased upon $NPY1$ gene deletion, whereas red bubbles show those with increased NAD-modification ratio. The bubble size indicates the extent of the relative change. **d** Npy1 in vitro kinetics of decapping different NAD-RNAs. NAD-RSM10 (5'-end fragment, 107 nt), NAD-LSM7 (5'-end fragment, 30 nt), and NAD-TDH3 (5'-end fragment, 99 nt) were assayed. All conditions as in Fig. 3a with the exception of a twofold higher enzyme concentration. Three independent experiments were performed, $n = 3$. **e** Quantification of Npy1-mediated 5'-NAD decapping over time, as shown in Fig. 4d. Error bars represent the mean ± SD. Three independent experiments were performed, $n = 3$. Source data are provided as a Source Data file.

NAD-RNA observed in every strain, and added a low-copy plasmid in which we inserted a DNA fragment containing the 600 bp upstream of the *TDH3* gene, containing the entire promoter region, plus the first 54 bp after the experimentally observed TSS of the TDH3 RNA (39 nt 5'-UTR, 15 nt coding sequence), followed by the ORF of *superfold-GFP* to monitor gene expression (Supplementary Fig. 5q). Mutants were prepared in which the Y at position −1, the A at position +2 and the G at position +3 were individually varied. An additional mutant was generated in which all A's in the tail region (+4, +5 and +9) were replaced. Cells were transfected with these plasmids, and harvested around $OD_{600} = 0.8$. Total RNA was isolated, treated with ADPRC, followed by click biotinylation, streptavidin purification, and RT. qPCR with gene-specific primers was used to assess the percentage of NAD-modified TDH3 RNA in each strain, using pure synthetic spike-in NAD-RNA and ppp-RNA, to ensure equal reactivity of each sample. This analysis revealed indeed strong (~2-fold) reduction of relative NAD incorporation

upon mutating positions −1 and +3, whereas for positions +2 and the A-rich tail the observed effects were not statistically significant (Fig. 7c and Supplementary Fig. 5r). Quantification of green fluorescent protein (GFP) expression levels revealed that mutating position −1 significantly decreases both NAD-RNA and non-NAD-RNA, whereas mutating position +3 modulates exclusively NAD-RNA (Fig. 7d and Supplementary Fig. 5s). Thus, a specific promoter sequence and particularly a G at position +3 are responsible for efficient NAD incorporation in vivo.

**Most NAD-RNAs are 3'-truncated**. The observation that most yeast RNAs enriched in NAD captureSeq are much shorter than full-length mRNAs and their preferential mapping to mRNA 5'-ends lead us to ask whether there is an influence of NAD incorporation on the transcript length. Globally, we determined the percentage of mRNA-mapped full-length reads for WT and

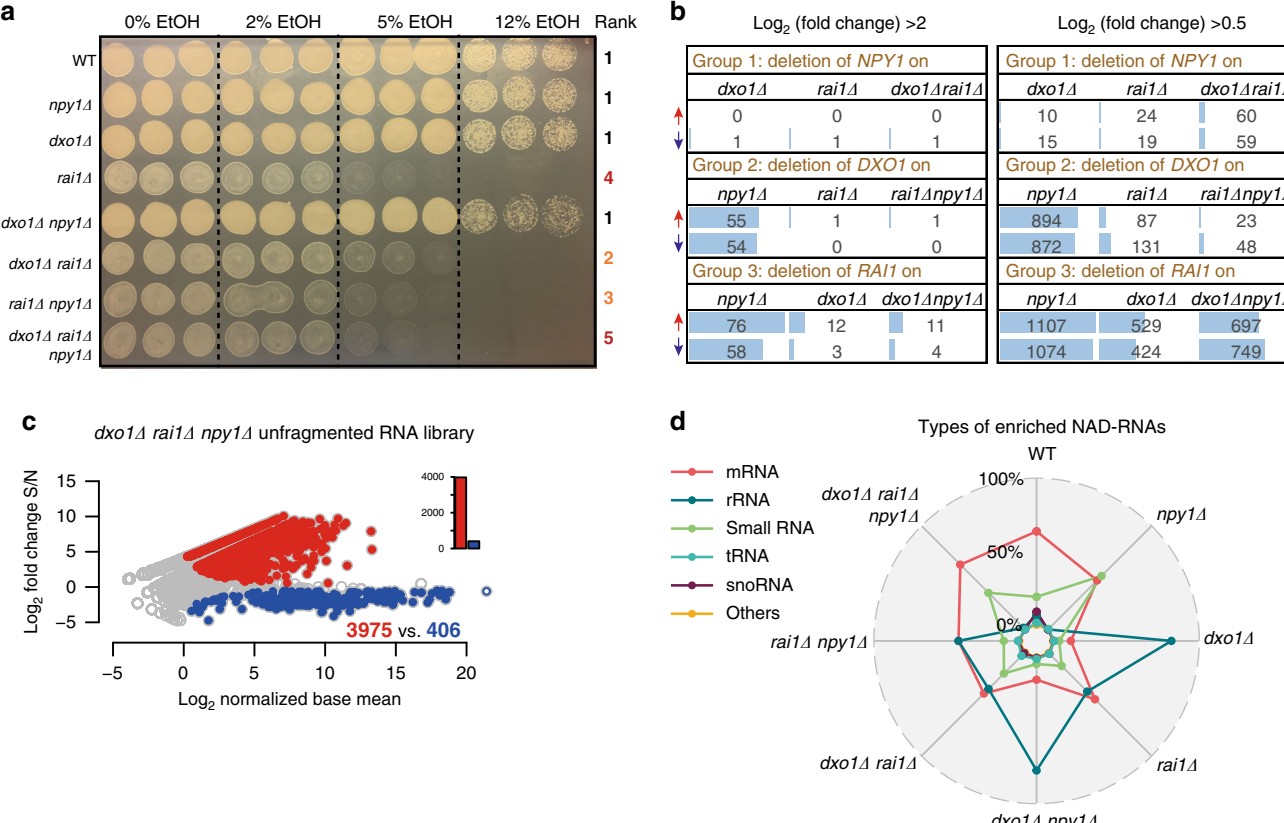

**Fig. 5 Enzyme deletions influence the NAD-RNA landscape. a** Growth phenotype of the WT and mutant yeast strains in the presence of different ethanol concentrations. The rank of approximate growth density is given on the right side. Each condition was tested in triplicates. Strain names are denoted on the left. **b** Effect of the deletion of one particular enzyme in different strains on the upregulation (red arrows) or downregulation (blue arrows) of RNA species, assessed by transcriptome sequencing. Group 1 summarizes the deletion of the *NPY1* gene from the single mutants (*dxo1Δ, rai1Δ*) and from the double mutant (*dxo1Δ rai1Δ*). Groups 2 and 3 show deletion of the *DXO1* gene and of the *RAI1* gene, respectively. The number of RNA species with log2FC > 2 is shown on the left panel, whereas with log2FC > 0.5 is shown on the right panel. **c** Enriched NAD-RNAs in the unfragmented NAD captureSeq library of the *dxo1Δ rai1Δ npy1Δ* strain. All parameters are as in Fig. 1a. Performed in biologically independent replicates, *n* = 3. **d** Radar plot of the distribution of different classes of enriched NAD-RNAs (in %) in different deletion strains. The colors indicate the type of RNAs, enumerated on the left. Source data are provided as a Source Data file.

all mutants and compared this value for sample (+ADPRC, S) and negative control (−ADPRC, N). After normalization to non-enriched species, in all eight libraries the sample group contained less full-length reads than the negative control and more abortive fragments (Supplementary Fig. 6a). At the individual transcript level, we determined for the highly expressed (and enriched) TDH3 RNA the transcript start and end nucleotide analyzing each read individually[26]. According to this analysis, both S and N groups feature the same dominating TSS (Fig. 8a, b, histogram on top), whereas a dramatically different abundance and size distribution of truncated 3′-ends was observed between S group and N group. The proportion of full-length Illumina reads with identical TSS differed by a factor of 2.7 (33.4% in S and 88.8% in N, Fig. 8a–b, histograms to the right of the two-dimensional plot). Although the reasons for this increased proportion of 3′-truncated NAD-RNAs remain unclear, these findings may suggest that unidentified quality control (QC) mechanisms detect NAD incorporation into RNA as an error quite early and interfere with efficient transcript elongation.

**Npy1, Dxo1, and Rai1 target different NAD-RNA populations.** The observation that the promoter motif got increasingly "blurry" with increasing number of decapping enzymes present (Fig. 7a and Supplementary Fig. 5e–k) supported our assumption that the

NAD captureSeq data actually reflect a superposition of RNAP II and decapping enzyme preferences. The comparison of the datasets of the three single mutants revealed extensive overlap, and 1544 species (>60%) were enriched in all three mutants, compared to the WT (Supplementary Fig. 6b). Similar findings were observed comparing the three double mutants. Computational sequence and secondary structure analysis of RNAs of uniquely or commonly enriched RNAs did not reveal specific features indicative of substrate preferences of these enzymes. However, for Rai1 we observed a slightly decreased minimum free energy of folding[27] for preferred RNA substrates, compared to poor ones (Supplementary Fig. 6c). This finding may suggest that Rai1 tends to have a preference for less structured 5′-ends.

We noticed that the removal of decapping enzymes not only influenced the number of RNA species enriched in NAD captureSeq and their enrichment values, but also the (apparent) length of their 5′-ends (TL). This phenomenon was observed for ~20% of all RNA species and occurred in both directions, namely (apparent) TL lengthening and shortening upon knockout. For example, among the 1100 enriched sequences in common between the WT and *npy1Δ* strain, 152 apparently got shorter and 75 got longer TLs (Fig. 8c). For all other mutants, similar observations were made. For several candidate RNAs, these length differences could be confirmed by qRT-PCR with the cDNA from the NAD captureSeq samples (Fig. 8d). This

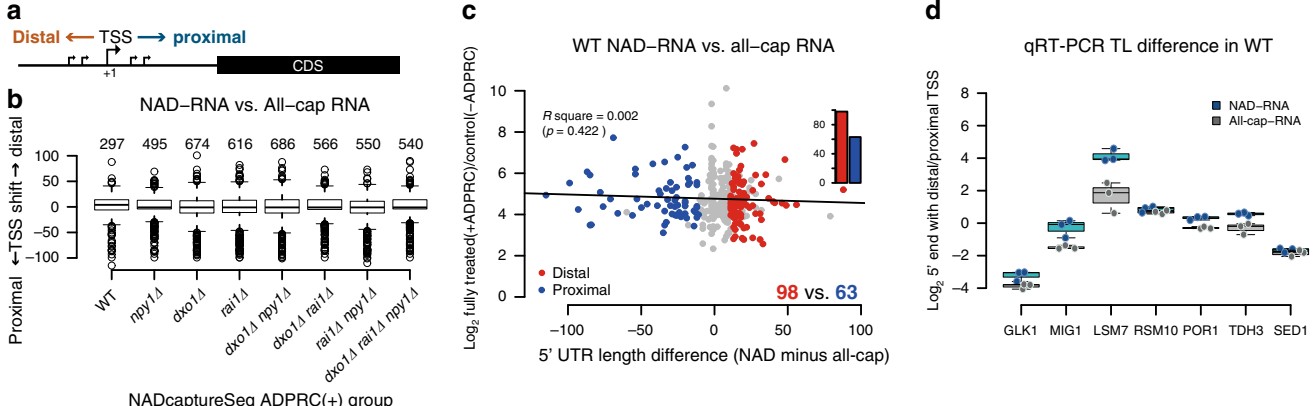

**Fig. 6 NAD-RNAs have longer 5′-UTRs than non-NAD-RNAs. a** Scheme of TSS shifting in proximal or distal direction. **b** Global TSS shifting between NAD-RNA (according to NAD captureSeq, unfragmented libraries) in all strains and canonical RNA (according to the dataset of Nagalakshmi et al.[18]). The boxplot shows from bottom to top minimum (Q1–1.5 interquartile range (IQR, 25–75%)), first quartile (Q1, 25%), median (solid line, 50%), third quartile (Q3, 75%), maximum (Q3 + 1.5 IQR), and outliers (black dots). The numbers above each box group indicates the number of RNA species analyzed in the corresponding strain. **c** Detailed TSS shifting between NAD-RNA and All-cap RNA in WT strain, 5′-UTR length difference denotes NAD-RNA 5′-UTR length minus canonical (all-cap) RNA 5′-UTR length. Red dots represent NAD-RNA species with distal TSS ( > 10 nt difference) and blue dots represent NAD-RNA species with proximal TSS ( > 10 nt difference). $FDR_{wilcox} < 0.1$. The sample size (*n*) is 347. The black line represents the linear correlation between 5′-UTR length and NAD-RNA enrichment level. **d** qRT-PCR with different primer pairs confirming different transcript leader (TL) lengths between NAD-capped RNAs (enriched fraction from NAD captureSeq library, blue boxes) and all-capped RNAs (from non-enriched input RNA, gray boxes). The boxplot displays the $\log_2$ of the ratio of the number of transcripts derived from the distal TSS to that from the proximal TSS. The layout of boxplot is as defined in **b**. Dots represent individual biological triplicate measurements. Source data are provided as a Source Data file.

phenomenon was observed almost exclusively for RNAs with read patterns indicative of pervasive transcription or multiple TSSs, and not for those with homogenous TLs. We assumed that the most likely explanation for these results may be that the decapping enzyme, when presented with a transcript mixture with different TLs, decaps some more rapidly than the others, due to sequence or structural preferences, thereby causing changes in the NAD captureSeq read profiles that look like shifted TSSs. A direct modulation of transcription (e.g., as transcription factors) is difficult to reconcile with the currently assumed roles and locations of these proteins, at least for Dxo1 and Npy1.

Rai1 has been reported as a nuclear protein and was detected as a component of the RNAP II elongation complex[28], whereas for Dxo1 both nuclear and cytosolic locations were claimed[29]. Npy1 was described as a peroxysomal protein[30]. Localization microscopy using three different C-SWAT fluorescent protein fusions[31] for each candidate gene revealed strong localized fluorescence in the nucleus for Rai1, whereas Dxo1 showed only a very weak and ubiquitous fluorescence (Fig. 9a), consistent with the reported localizations of these enzymes. For Npy1, however, a rather homogenous cellular distribution without enrichment at specific sites was observed, consistent with cytosolic localization (Fig. 9a). This localization may imply a temporal order, in which Rai1 processes its NAD-RNA substrates during or shortly after transcription, whereas Npy1 can only act once the transcripts (or their primary degradation products) arrive in the cytosol. For Dxo1, both options are conceivable. Therefore, we tried to find evidence in our NAD captureSeq data for a temporal order of processing by these enzymes. In particular, we searched for examples where—when starting with the triple knockout and then "adding" one by one the decapping enzymes (i.e., comparing the triple knockout with the appropriate double and single knockouts)—a significant TL length change is observed upon "addition" of the first decapping enzyme (suggesting that this enzyme decaps a fraction but not all TL variants) and upon "addition" of the second one the transcript disappears entirely (or is significantly reduced in enrichment) from the enriched fraction (suggesting that the second enzyme decaps the remaining TLs).

Indeed, from the 84 species with TL length changes between triple knockout and *dxo1Δ npy1Δ* double knockout, 61 disappeared in the *npy1Δ* single knockout and 38 in the *dxo1Δ* single knockout. Importantly, hardly any examples were found for the pathways via the other double mutants (one example for *npy1Δ rai1Δ* and 0 for *dxo1Δ rai1Δ*) (Fig. 9b). These findings are consistent with our assumption that Rai1 is the first factor in NAD-RNA decapping.

**NAD-RNAs are not translatable in vitro.** Finally, we tested whether the NAD cap in combination with different TL lengths and sequences may modulate translation. Reports on NAD-RNA translatability are conflicting: Jiao et al.[7] had reported that NAD-RNA is not translated in human (HEK293T) cell extracts, based on a single mRNA luciferase construct with a single fixed TL sequence, while a recent study in the model plant *Arabidopsis thaliana* demonstrated that NAD-capped mRNAs are enriched in the polysomal fraction, associate with translating ribosomes, and can probably be translated[9]. No data for yeast have been reported yet. For seven different mRNAs, we prepared luciferase fusion constructs with long and short TLs by in vitro transcription, followed by the removal of the accompanying ppp-RNA by treatment with polyphosphatase and exonuclease Xrn-1. Although the control constructs harboring an m7G-capped 5′-end were efficiently translated in a yeast in vitro extract and showed significant differences in luminescence depending on the TL length[19,20], NAD-capped RNA was not translated to any significant extent, even less than ppp-RNA and p-RNA of the same sequence (Fig. 10a). These results suggest that NAD-capped RNAs (at least the nuclear transcripts investigated here) are not translated in budding yeast.

## Discussion

Taken together, our results indicate that in budding yeast, NADylation of RNAs is a very common phenomenon. A previous study reported only 37 species enriched in NAD captureSeq in budding yeast grown in the same medium[8]. This study, however, focused on full-length mRNAs and used a library preparation

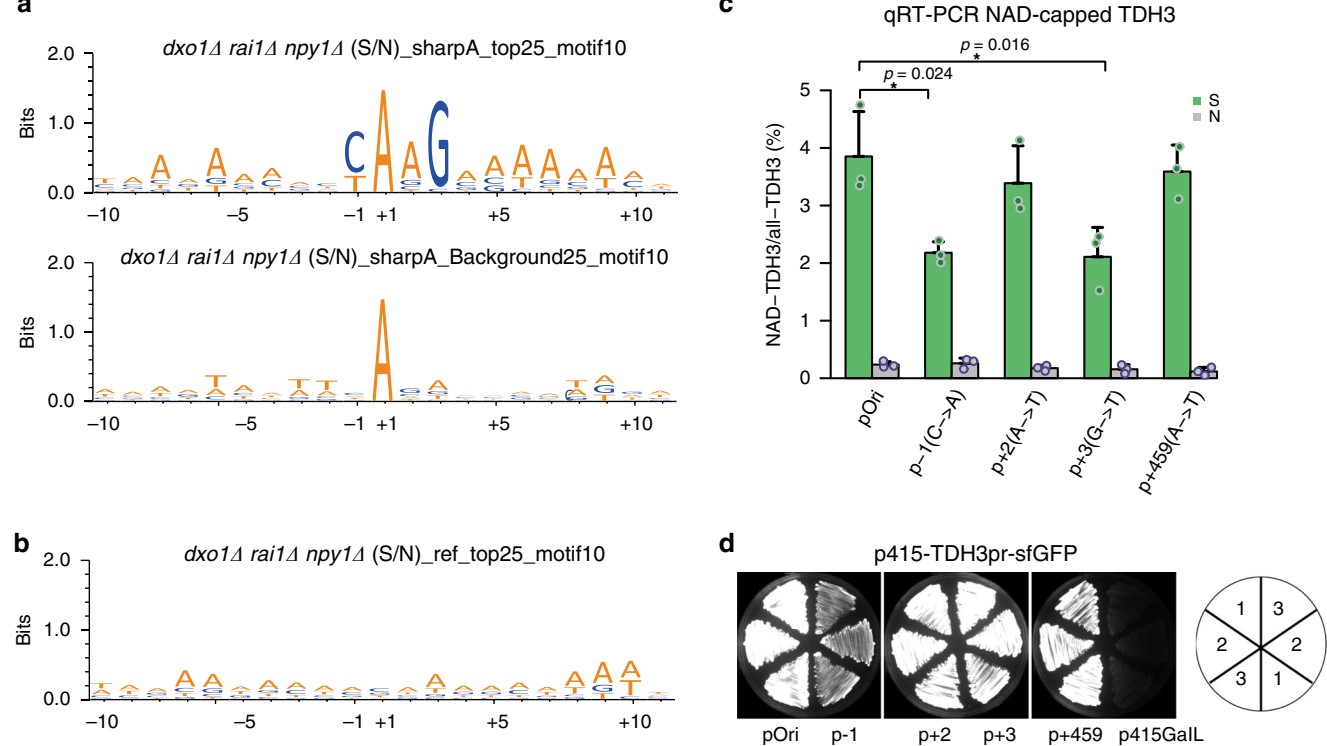

**Fig. 7 NAD incorporation by RNAP II is promoter dependent. a** Motif analysis of the −10 to +10 region around the TSS based on NAD captureSeq data. Top25 and background25 represent the 25 NAD-mRNA species with the highest enrichment values and the 25 most abundant, but not significantly enriched mRNA species ($0.707 \leq S/N \leq 1.414$) with a *sharp A* feature at position +1 (TSS) in the *dxo1Δ rai1Δ npy1Δ* strain. S/N is the enrichment of NAD-RNA in the sample group (+ADPRC) compared to the negative control group (−ADPRC). *Sharp A* means an "A" at position +1 with a more than three-fold higher signal than that of the nucleotide in the −1 position. The letters, representing the four nucleobases, were colored dark orange for A/T and blue for G/C. **b** Motif analysis of the same Top25 enriched RNA species, using the canonical TSS from Nagalakshmi dataset[18]. **c** Quantification of the NAD-modification ratio using the *TDH3* gene promoter and the relevant mutant promoter sequences in vivo. The height of the green bar indicates the TDH3 RNA NAD-modification ratio in the sample group (+ADPRC), whereas the gray bar denotes the same in the N group (−ADPRC). The pOri represents the original TDH3 promoter and promoter mutants are enumerated below the bar chart. Dots represent biological triplicates and error bars represent SDs. *p*-values are denoted by asterisks: \**p* < 0.05 (Student's *t*-test, one sided). **d** Expression of sfGFP under a *TDH3* gene promoter or relevant promoter mutants in vivo. pOri, $p − 1$, $p + 2$, $p + 3$, and $p + 459$ are the same as in Fig. 7c. p415GalL is negative control strain for background. All strains were cultured in biological triplicates, $n = 3$. Source data are provided as a Source Data file.

protocol that discarded the small RNA fraction (≲200 nt). Our work confirms that there are hardly any full-length NAD-mRNAs, but additionally reveals a rich landscape of thousands of short NAD-mRNA fragments whose purpose is apparently not to encode for proteins.

This conclusion may not apply in yeast mitochondria, however, where several lines of evidence suggest that transcriptional incorporation of the coenzyme is, in fact, an evolved feature. First, the mitochondrial transcription machinery exhibits NAD-mediated RNA initiation efficiencies that are at least tenfold higher, compared to the nuclear RNAP II[32]. Second, individual mRNA species in this organelle were found to be highly 5′-NAD modified, comprising up to 60% of the respective transcript pools. Third, the redox state of mitochondrial NAD caps was observed to vary, depending on the metabolic growth conditions of the cell[32]. These findings may indeed indicate a regulatory role of the NAD cap in this organelle, which harbors redox-intensive energy conversion pathways highly dependent upon the coenzyme, and are contrasting sharply with our discovery of a tightly policed landscape of nuclear-derived NAD-mRNA fragments.

In addition, the identity of a dedicated surveillance machinery within the mitochondrial matrix is of considerable interest. The three decapping enzymes investigated here, however, are unlikely to encounter mitochondrial transcripts, as they lack the required

targeting sequence and are absent in yeast mitochondrial proteomic data[33,34].

NAD is initially incorporated into several thousands of transcripts by transcription initiation by RNAP II in a largely statistical manner reflecting the competition of NAD with the canonical initiator ATP. As in prokaryotes[5,35,36], the promoter sequence determines the efficiency of NAD incorporation, which is for most yeast nuclear transcripts between 1% and 5%. We observe that a YAAG motif supports efficient NAD incorporation by RNAP II in vivo, with the G at position 3 being particularly important. This finding does not rule out the existence of unknown alternative post-transcriptional pathways for NAD incorporation, e.g., for enriched snoRNAs or rRNAs, or for enriched mRNAs without the YAAG motif. In contrast to the promoter requirements determined here, a preference for HRRASWW was reported for *E. coli* RNAP[36], WARR for *B. subtilis* RNAP[5], and RA for yeast mtRNAP[32], with the underlined A always indicating the TSS. It should, however, be noted that the *E. coli* RNAP and yeast mtRNAP consensus motifs were established using an entirely different methodology, making a direct comparison difficult. We find that, as a consequence of this promoter dependence, for many RNAs the NADylated species originate from different TSSs and have therefore different (shorter or longer) 5′-UTRs than the canonical ones. This phenomenon may modulate the secondary structure of these RNAs

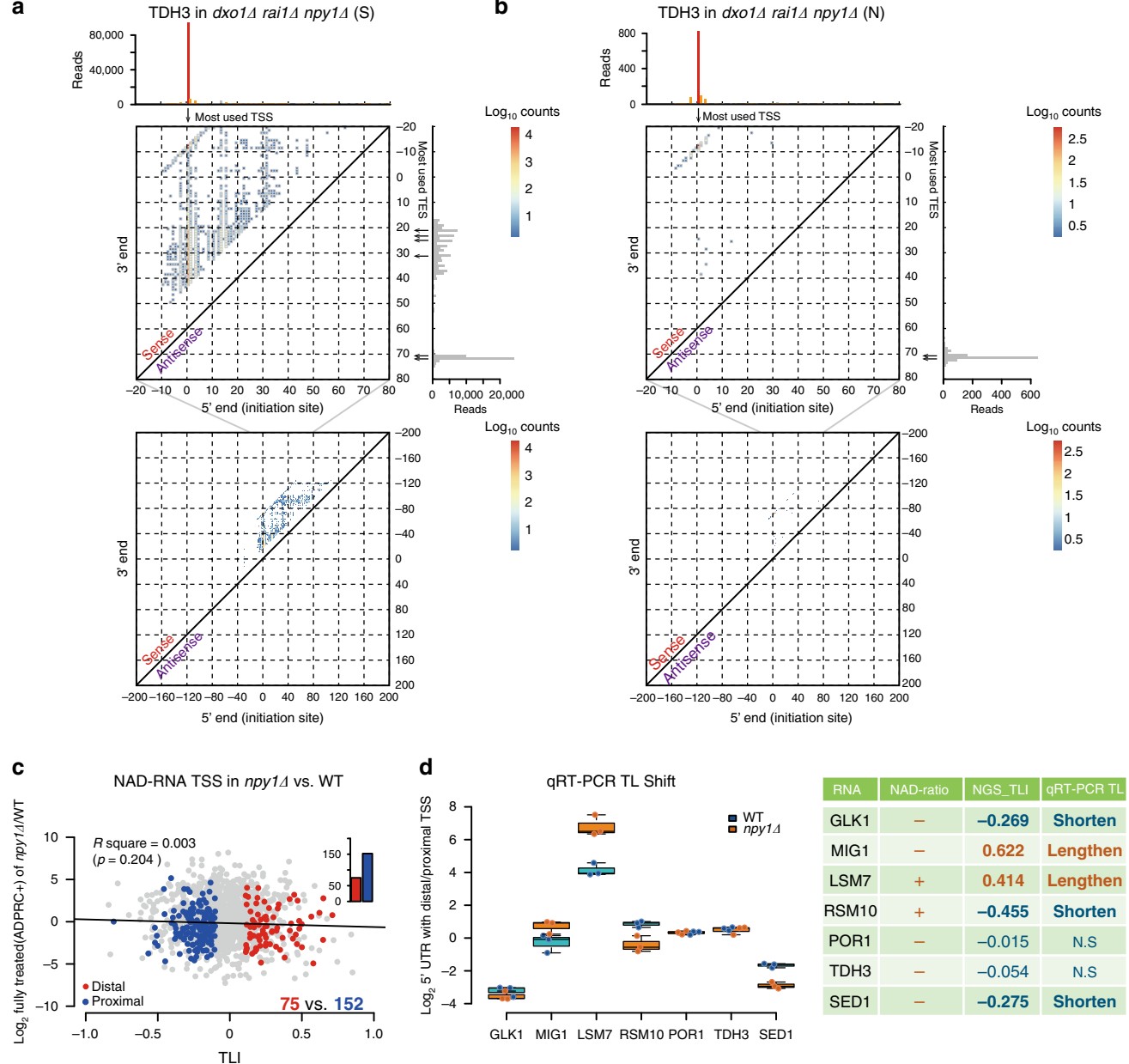

**Fig. 8 The 5′- and 3′- end heterogeneity of NAD-RNA. a**, **b** 2D NAD-capping single transcript plot for TDH3 RNA in the *dxo1Δ rai1Δ npy1Δ* strain for the sample group (+ADPRC, S, (**a**)) and the N group (−ADPRC, N, (**b**)). 0 denotes the RefSeq TSS (−39 nt of the TLS site). Each bin represents a unique 5′ (initiation site, TSS, *x* axis) and 3′ (transcript end site, TES, *y* axis) pairing colored by the number of reads mapped to that bin. Expanded view below. For the 5′-end initiation site histogram (above the 2D plot), the red bar indicates the "A" position with YAAG feature, whereas orange bars indicate "A's" without this feature and gray bars indicate U/C/G. For the 3′-end histogram, no color differentiation was performed. Arrows represent preferred TSSs in the 5′-dimension and TESs in the 3′-dimension. **c** Global effect of the deletion of the *NPY1* gene on the transcript leader length by comparing NAD captureSeq read starts in the *npy1Δ* mutant with the WT (unfragmented libraries). A Transcript Leader length Index (TLI) > 0 indicates that upon *npy1Δ* mutation the TSS towards more distal positions, while a TLI < 0 indicates a proximal shift; RNAs with significantly (FDR < 0.05) shifted TSS are shown as colored dots. Red: distal shift (TLI > 0.1) Blue: proximal shift (TLI < −0.1). The y axis indicates the relative NAD-RNA enrichment difference (ratio) from NAD captureSeq libraries between the two strains. The bar charts represent the total number of RNA species with distal (red) or proximal (blue) TSS shift. **d** qRT-PCR with different primer pairs confirming different NAD-RNA transcript leader (TL) lengths in the enriched fractions from NAD captureSeq libraries between WT (blue boxes) and *npy1Δ* strain (orange boxes). All parameters and the layout of boxplot are as in Fig. 6d. In the table, symbol "−" and "+" indicate a decrease or increase of the NAD-modification ratio, respectively. "NS" indicates no significant difference. Source data are provided as a Source Data file.

and hence their stability, molecular interactions, and biological fate. Of note, the discovery of alternative TSS selection and the YAAG core promoter motif have been made possible by the combination of 5′-end selection by ADPRC treatment and ligation-based attachment of the 5′-adapter, which allowed determination of NAD-RNA 5′-ends with single-nucleotide

precision, in contrast to random-primed library preparation methods that create heterogeneous ends. An increased affinity of RNAP II for the YAAG motif, while the polymerase transiently harbors NAD in its catalytic site, could potentially also explain the enrichment of non-canonical TSSs upon execution of NAD captureSeq. An in-depth biochemical investigation should explore

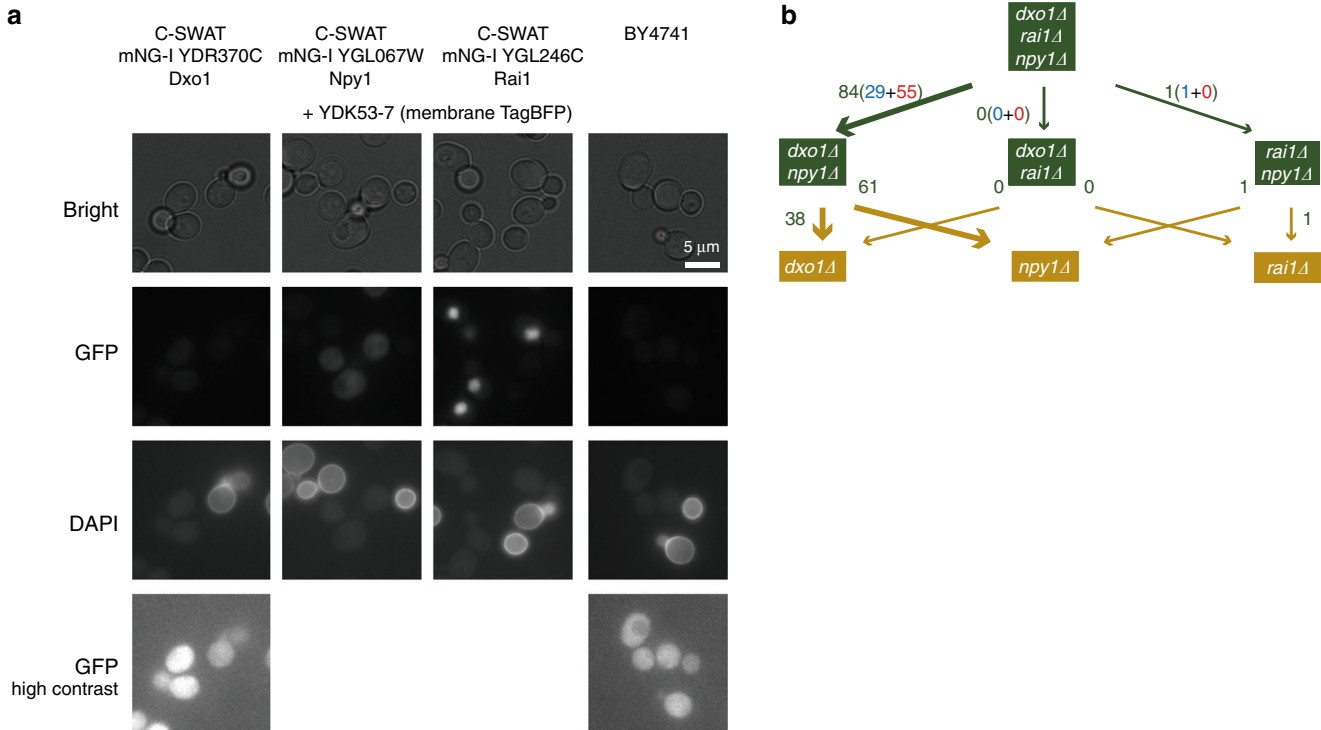

**Fig. 9 Decapping enzyme localization and hierarchy. a** Subcellular localization of Npy1, Dxo1, and Rai1. The proteins of interest were visualized as C-SWAT NeonI fusion proteins. The membrane marker protein TagBFP served as background control and was visualized in the DAPI channel. Due to the low intensity of the GFP signal for Dxo1, this image is shown again in high contrast, as is the BY4741 WT strain, exhibiting autofluorescence. Representative micrographs of three biologically independent replicates are shown, $n = 3$. **b** Analysis showing the number of RNA species that exhibit a TL shift upon reintroduction of the first decapping enzyme (upper part, numbers next to the green arrows) into the $dxo1\Delta$ $rai1\Delta$ $npy1\Delta$ deletion background, and the number of those disappearing entirely or showing reduced enrichment upon introduction of the second enzyme (lower part, numbers next to yellow arrows. TL shifting is defined by TLI > 0.1 or < −0.1 and FDR < 0.1. Blue and red numbers represent TL shifting to the proximal and distal side, respectively. For example, the route from the $dxo1\Delta$ $rai1\Delta$ $npy1\Delta$ triple knockout to the $dxo1\Delta$ $npy1\Delta$ double knockout (i.e., introduction of Rai1) identifies 84 RNA species with shifted TLs, 38 of which disappear or show reduced enrichment upon additional introduction of Npy1 (tracing the path from the $dxo1\Delta$ $npy1\Delta$ to the $dxo1\Delta$ mutant). Source data are provided as a Source Data file.

the possibility of NCIN-mediated guidance of RNAP II, and other RNAPs, to distinct TSSs.

NAD-RNAs are—on average—shorter than non-NAD-RNAs and only rarely reach the size of a typical primary mRNA transcript. The most likely explanation is that some unidentified QC mechanism detects 5′-NADylation of RNA as an error early during transcription and prevents efficient elongation, as it does with uncapped or incompletely capped transcripts[37]. Alternatively, NAD-RNAs might be subject to accelerated degradation after transcription is complete, but it is unclear how 5′-NAD can accelerate degradation at the 3′-end.

The discovery that budding yeast maintains at least three different, partly redundant, pathways for NAD cap removal, using enzymes with different chemistry and cellular localization, implies that decapping unwanted NAD-RNAs is important for the cell. Our data are in agreement with the hypothesis that Rai1 acts earlier than the other two enzymes. As the nuclear protein Rai1 is known to associate with RNAP II during elongation[28] and to act in RNA surveillance by assisting the 5′- to 3′- exonuclease Rat1 in the co-transcriptional degradation of uncapped transcripts[38,39], such an order appears plausible. The Rat1-Rai1 complex is hereby believed to play an important role by mediating 5′-end cap QC (5′-QC) in the yeast RNAP II transcription cycle, following the transcription checkpoint pause stage, whereby RNAP II enters transcription elongation upon phosphorylation of distinct serines within the C-terminal repeat domain of the polymerase[40]. Surveillance and hydrolysis of the accidentally incorporated 5′-NAD cap could be enacted in a mechanistically

similar manner, mirroring the clearance of unmethylated, aberrantly capped mRNAs by the Rat1-Rai1 heterodimer[38,41].

An earlier report, providing evidence that dinucleotide hydrolysis mediated by *Sachharomyces cerevisiae* Npy1 is not entirely restricted to NAD, but also includes the redox cofactor flavin adenine dinucleotide among others[30], warrants a thorough biochemical characterization to define the set of RNA 5′-metabolite caps, targeted by this enzyme. A corresponding study should hereby follow the example set by the systematic and meticulous elucidation of RNA 5′-cap specificities of mammalian DXO and *Schizosaccharomyces pombe* Rai1[42,43].

The observed combination of the low efficiency of RNAP II transcription initiation by NAD, the reduced length of NAD-RNAs, and the abundance of NAD-RNA decapping enzymes warrants that hardly any NAD-RNAs occur in the cell that could give rise to translation into proteins. Our data indicate, however, that yeast ribosomes, such as mammalian ones[7], hardly translate synthetic NAD-mRNAs, suggesting that the ribosomal machinery contains additional safeguards against NAD-mRNAs. Thus, budding yeast protects itself at different stages of gene expression against NAD-RNA.

## Methods

**Yeast strains.** Unless otherwise stated, *S. cerevisiae* strains were grown in yeast extract/peptone/dextrose media (YPD). All strains used in this study except of YDK587-1 and its derivatives, YDK53-7, and C-SWAT mNeonGreen (mNG-I) strains were derivatives of the S288C strain BY4742 (*MATα his3Δ1 leu2Δ0 lys2Δ0 ura3Δ0*) and are listed in the key resource table. YDK53-7 and C-SWAT mNeonGreen (mNG-I) strains were derivatives of the S288C strain BY4741 (*MATa*

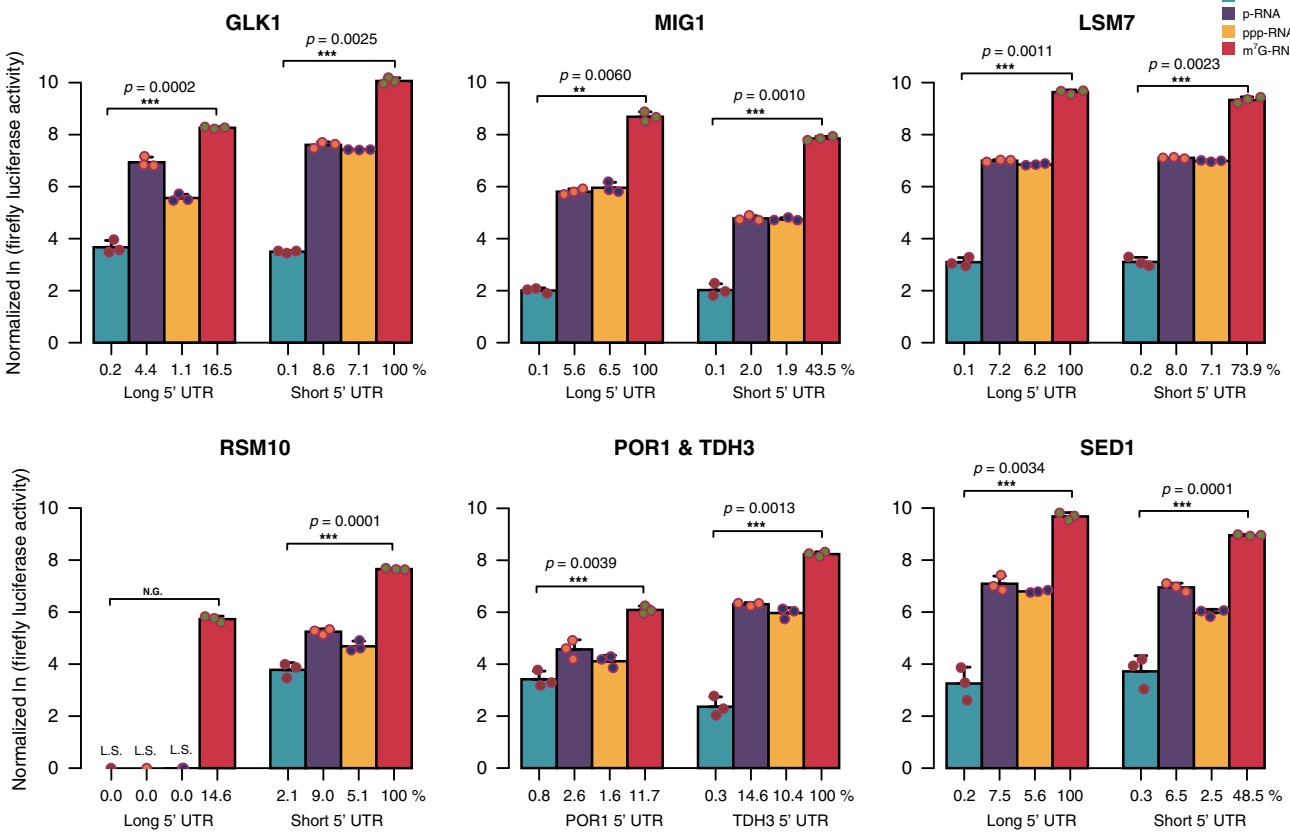

**Fig. 10 In vitro translation of NAD-RNA with shifted TLs.** In vitro translation of NAD- RNA, p-RNA, ppp-RNA, and m7G-RNA with short or long TL sequences. Probed mRNAs, bearing a 5′-NAD-, 5′-p-, or 5′-ppp-terminus, contain the alternative TL sequences identified in NAD captureSeq, followed by 22 nt of the CDS of the corresponding gene, followed by the firefly luciferase CDS (1653 nt) and a poly(A)$_{30}$ tail. Reference m7G-capped mRNA contains *Renilla* luciferase CDS (936 nt) and a poly(A)$_{30}$ tail. Firefly luciferase activity was normalized to *Renilla* luciferase activity, then normalized to the $C_t$-value of the full-length mRNA determined by qRT-PCR. NAD-capped mRNAs, p-mRNAs, and ppp-mRNAs contain the same sequence as the corresponding m7G-capped RNAs. Logarithmic representation of normalized luciferase activity. Blue, NAD-RNA; purple, p-RNA; orange, ppp-RNA; red, m7G-RNA. The percent values below the bars indicate relative luciferase activity normalized to the m7G-capped mRNA of that species with the higher luciferase expression. L.S. denotes that the luciferase signal was not significantly above the background. Error bars represent mean + SD. Three independent experiments were performed, $n = 3$. $p$-values are denoted by asterisks: *$p < 0.05$; **$p < 0.01$; ***$p < 0.005$ (Student's $t$-test, NAD-RNA vs. m7G-RNA, one sided). Source data are provided as a Source Data file.

his3Δ1 leu2Δ0 met15Δ0 ura3Δ0) and YDK587-1 was a derivate of the S288C strain ESM356-1 (MATa ura3-52 leu2Δ1 his3Δ200 trp1Δ63). Strains were cultivated in YPD medium at 30 °C according to standard protocols. Antibiotics were used at the following final concentrations: 200 μg mL⁻¹ geneticin (G418), 300 μg mL⁻¹ hygromycin B, and 100 μg mL⁻¹ nourseothricin. Gene deletions were performed using standard PCR-based recombination methods as described[44,45], followed by PCR-based confirmation. Single, double, and triple mutants were generated by mating and sporulation followed by random spore isolation. Plasmid transformations were performed using standard methods[46,47] and transformants were selected on synthetic complete (SC) medium lacking leucine for p415-based plasmids.

**Total RNA isolation and purification.** Total RNA from yeast BY4742 strains was isolated by the hot phenol method[48] with minor changes. Cells were collected (OD$_{600}$ 0.8) from 0.5 L YPD medium, quickly frozen in liquid nitrogen, and stored at −80 °C until all samples were ready. Cell pellets were thawed on ice and washed with dH₂O, resuspended in 8 mL TES solution (10 mM Tris-HCl pH 7.5, 10 mM EDTA, 0.5% SDS) and were added to 8 mL phenol. The mixture was vortexed thoroughly for 1 min, incubated for 60 min at 65 °C with occasional shaking (50 s per 10 min, 550 r.p.m.), and placed on ice for 15–20 min. Then the samples were centrifuged at 14,000 × $g$, 4 °C. The aqueous supernatant was subjected to phenol extraction, P/C/I purification, and chloroform purification. RNA was precipitated with 0.1 volumes 3 M NaOAc pH 5.5 and 2.5 volumes ethanol at −20 °C overnight. Precipitated RNA was dissolved in 2 mL dH₂O. RNA concentration was determined by Nanodrop spectrometry and its integrity analyzed by 1.2% formaldehyde-denaturing agarose gel electrophoresis. RNA (1 mg) was treated with 100 U DNase I in 1X DNase I buffer (Roche) for 40 min at 37 °C. DNase I was removed by P/C/I extraction (twice), followed by ethanol precipitation of the RNA

(−20 °C, overnight). The pellet was dissolved in 100 μL dH₂O, the RNA concentration determined by Nanodrop, and its integrity analyzed by both denaturing agarose gel electrophoresis and Bioanalyzer RNA integrity analysis.

**NAD captureSeq and transcriptome libraries preparation.** Standard, unfragmented NAD captureSeq: Total RNA was subjected to the NAD captureSeq protocol[11] with minor modifications, as outlined in the following. Total RNA, obtained in biological triplicates, was used as starting material for library preparation. More specifically, 100 μg total RNA was commonly supplemented with 5 ng NAD-RNAI ("spike-in" control) and treated with ADPRC (Lab stock, 22.5 U), in the presence of 10 μL 4-pentyn-1-ol at 37 °C and for 60 min in 100 μL total reaction volume, including 50 mM Na-HEPES pH 7.0, 5 mM MgCl₂. The same reaction mixture, omitting the ADPRC enzyme, served as background control (negative control). The reactions were stopped by adding 100 μL dH₂O and 200 μL P/C/I reagent. This was followed by performing P/C/I extraction twice and three additional ether extractions. Subsequent copper-click reactions, captured by streptavidin bead, preadenylated-3′-adapter ligation, RT, Cytidine triphosphate(CTP)-tailing, anchored-5′-adapter ligation, and cDNA PCR amplification with barcode oligos were performed as standard library preparation[11]. Library QC was conducted by Sanger sequencing. Specifically, the amplicon derived from the NAD-RNAI spike-in control and other cDNA amplicons were amplified using corresponding adapter-ligated sequences. Resulting cDNA libraries were then cloned into plasmid(s) for Sanger sequencing. After the successful pre-sequencing, PCR products were purified and size-selected within a range of 150–300 bp by 10% native polyacrylamide gel electrophoresis (PAGE). The absence of primer-dimers was ensured by Bioanalyzer 2100 (Agilent) analysis using the Agilent High Sensitivity DNA Kit. Concentrations were measured using the Qubit approach. Primer-dimer-free libraries were then multiplexed at a final concentration of ~20 nM. Based on the cDNA length

distribution, the library pools were either supplemented with 20% v3 Phix Control (Illumina) or custom Illumina sequencing primers, which bear three Gs at their 5′-end, preceded by the Illumina standard sequencing, to mitigate library imbalances before NextSeq 500 75 bp single-end (SE) sequencing.

Fragmented NAD captureSeq: Biological triplicates of total RNA (gDNA free) were used as library starting material. Total RNA (100 μg), supplemented with 5 ng NAD-RNAI (as optional spike-in control), were randomly sheared in a 65 μL reaction volume that contained 32.5 μL alkaline fragmentation solution (2 mM EDTA, 10 mM Na$_2$CO$_3$, 90 mM NaHCO$_3$ pH 9.3) at 94 °C (5 min for WT strain and 20 min for $npy1\Delta$ strain to approach similar fragment size). Sheared RNA fragments were visualized on a 1.2% formaldehyde-denaturing agarose gel. Next, the sheared RNA was precipitated, in a total volume of 200 μL dH$_2$O, by addition of 600 μL ethanol, 20 μL 3 M NaOAc pH 5.5, and 1 μL glycogen at −20 °C overnight. The precipitated RNA was washed with 75% ethanol and subsequently dissolved in 50 μL dH$_2$O. An equivalent of 100 μg sheared RNA was then treated with 100 U T4 PNK, along with 0.1 mM ATP, 100 mM imidazole-HCl pH 6.0, 10 mM MgCl$_2$, 10 mM β-mercaptoethanol, and 20 μg ml$^{-1}$ RNase-free bovine serum albumin (BSA) in a total volume of 200 μL at 37 °C for 5.5 h. RNA extraction was performed twice, employing the P/C/I approach, and followed by triple ether extraction and ethanol precipitation. Precipitated RNA was the washed again, using 75% ethanol, and ultimately dissolved in 20 μL dH$_2$O, yielding the library input for the standard NAD captureSeq protocol[11]. PCR products were size-selected within a range of 150 to 300 bp (referred to as "small fragmented NAD captureSeq library") and of 300–500 bp (referred to as "large fragmented NAD captureSeq library"), enabled by 10% native PAGE. Bioanalyzer QC, library multiplexing, and the overall sequencing strategy were executed in a similar manner, as done for the unfragmented NAD captureSeq. Again, the NextSeq 500 75 bp SE approach was chosen for sequencing.

Transcriptome libraries: Biological triplicates of total RNA (gDNA free) served as library input material. Total RNA (1 μg) was subjected to rRNA depletion by Ribo-Zero rRNA Removal Kit (yeast). rRNA-depleted RNA was randomly sheared in 10 μL dH$_2$O, at 94 °C 10 min. Fragmented RNA was processed using the NEBNext Ultra II Directional RNA Library Prep Kit for Illumina, following the instructions of the manufacturer. cDNA was barcoded by PCR amplification using NEBNext Multiplex Oligos for Illumina. Further cDNA size selection in the range of 300–500 bp was performed, employing the Agencourt RNAClean XP kit. Primer-depleted cDNA was examined by Bioanalyzer and the concentration was measured by Qubit. Multiplexed libraries were sequenced by NextSeq 500 75 bp SE.

**NGS analysis**. Unfragmented NAD captureSeq analysis: Original reads were demultiplexed, based on the PCR barcode, not allowing for mismatches and subjected to standard QC procedures. Further 5′-end leading (G/N)$_n$ and 3′-end adapters (C NNNNNN AGATCG) were trimmed (minimum length 12 nt) by in-house scripts. Reads that mapped to the spike-in internal standard (IS) RNAI sequence (bowtie -v 2, version 1.1.2) were then counted first. IS unmapped reads were subsequently classified as "small RNA reads" (12–17 nt) or "normal RNA reads" (>18 nt).

Normal RNA reads were mapped to the reference genome *S. cerevisae* BY4742 strain (BY4742_Toronto_2012, SGD) (bowtie -v 2). Normal RNA reads that mapped to rRNA genes or tRNA genes were separated and counted individually. Remaining reads that could not be mapped to rRNA genes were remapped to the yeast reference genome (bowtie --best --strata -M 1 -m 20 -v 2). The.sam files were then converted to.bam files and sorted (samtools, version 0.1.13). The.bam files were used to generate.wig files, which could be normalized by reads per million mapped reads, for single-nucleotide resolution-based analysis. Sorted.sam files were further filtered by strand-specific selection and used for three different types of analysis as follows:

On the one side, the filter-passed reads from sorted.sam files were used for linear trend tests[49] to perform an alternative TSS study. Briefly, the position of the first nucleotide of the mapped reads within the range of the 5′-UTR[18] with additional 50 nt upstream extension from each gene was collected. A sliding window of 18 nt was applied for searching TSS clusters from the highest density one to the lowest one. Then a 2 × n table was made, whereby n is the number of unique TSS clusters found in the sample group (S, +ADPRC) and/or the negative control group (N, −ADPRC). Values in the table represent the number of reads for these clusters in the S or N group (normalized by the proportion of the respective cluster relative to the sum of reads for all clusters), and are sorted from the shortest (left) to the longest UTR (right). For covariance analysis, the top row (S) was assigned a weight value of 2, whereas the bottom row (N) was assigned 1. Column (Y, n columns) weight was set to be the 5′-UTR length of each cluster. The Pearson's correlation r was calculated by $cov(X,Y)/(\delta_x \times \delta_y)$. The operator M was calculated based on the equation $M^2 = (n − 1)\, r^2$ ($n \geq 30$), following the $\chi^2$-distribution (freedom 1). The corresponding p-value was used for FDR calculation, employing the Benjamini–Hochberg method.

On the other side, the filter-passed reads were counted for RNA hits (htseq-count -m intersection-nonempty, version 0.6.0) based on the annotation.gff3 file (BY4742_Toronto_2012, SGD, mRNA regions were annotated by their 5′-UTR region (−120 to +65, translation starting site referred to as "0," described in Supplementary Fig. 1)). Raw hits were analyzed by DESeq2 (version 1.4.5, relevant

package from Bioconductor version 2.14) for NAD-RNA enrichment statistics. Meanwhile, raw hits were normalized by transcripts per kilobase million (TPM) for NAD-ratio calculation. Simply put, the relative total number of NAD-RNA (k) was the signal intensity from the ultra performance liquid chromatography (UPLC)/MS measurement of total RNA. The distribution of all-cap-RNA in total RNA was simulated by their TPM in transcriptome data. The distribution of NAD-RNA was simulated by their TPM in NAD captureSeq data. The relative individual NAD-ratio was calculated by $k*\mathrm{TPM_{NAD\text{-}RNA}}/\mathrm{TPM_{all\text{-}cap\text{-}RNA}}$.

In addition, the filter-passed reads, which mapped to 5′-UTRs of mRNAs, were collected for *sharp A* promoter motif analysis. This analysis included the genome-mapped position of the first base of reads referenced to the TSS site. For each individual mRNA, either the unique most abundant mapped position of the first base of reads or one of most abundant mapped positions (several ones with equal abundance) was defined as TSS (reference as +1) but this position was annotated as "A" in the genome. The sequence flanking the TSS from the −10 to +10 position, within the reference genome, was further analyzed for a consensus motif. The sharp value was defined as the ratio between the number of nucleotides accumulating at the +1 position and that at the −1 position, revealed by the generated.wig file. The sharpA feature was screened, based on the premise that the TSS constitutes an A and possesses a sharp value bigger than 4.

For the small RNA reads group, reads were purged by additional genome mapping (bowtie -v 0). Next, small RNA clusters mapping to mRNA 5′-UTRs were assembled together (60% sequence similarity, same strand direction, and max fold copy number difference is 50). Differential abundance of small RNA reads was calculated by the enrichment of reads in S group to N groups.

Fragmented NAD captureSeq was analyzed in the same way, as done for the unfragmented NAD captureSeq library.

Transcriptome analysis: Raw reads were demultiplexed and QC was performed using the same parameters, already employed for unfragmented NAD captureSeq analysis. Reads with 5′-/3′-end adapters were properly trimmed, leveraging an in-house script. Then the sequence of raw reads was converted into its reverse complement. Reads were then mapped to the yeast reference genome (bowtie -v 2). rRNA genes and tRNA gene reads that mapped independently, as well as rRNA-free hits counting (same annotation region on 5′-UTR of mRNA), were treated in a similar way as done in the NAD captureSeq analysis, described above. Also, .wig files and differential expression statistics were generated and performed according to the same procedure.

Function clusters were generated by David (version 6.7) and FGNet (version 3.16.0). Enriched promoter motifs were analyzed by MEME (version 5.0.3) and WebLogo (version 3). Meta-analysis of function clusters and pathways was enabled by Metascape (metascape.org) and cytoscape (version 3.7.0).

**RNA pulldown and UPLC-MS analysis**. Streptavidin Sepharose High Performance beads (250 μL) were loaded on Mobicol Classic columns. The column was washed three times with 1× phosphate-buffered saline (PBS) buffer, then five times 25 μL of 25 μM biotin-DNA probe (Biomers, Supplementary Table 1) were added sequentially. The mixture was incubated at 25 °C for 10 min. Next, the column was washed two times with 300 μL 1× PBS, followed by equilibration in 300 μL pulldown buffer (10 mM Tris-HCl pH 7.8, 0.9 M tetramethylammoniumchloride, 0.1 M EDTA pH 8.0). Total RNA (gDNA free; 200–500 μg) was added into the column and incubated at 65 °C for 10 min and then rotated (Tube Rotator, VMR) at 20 °C for 25 min. Next, the column was washed six times with 200 μL dH$_2$O, to remove unspecifically binding RNAs. RNA was eluted by adding four times 200 μL 2 mM EDTA (75 °C, pre-heated) under 10 s per min shaking (350 r.p.m.) at 75 °C for 10 min. The eluate was precipitated with 0.5 M ammonium acetate pH 5.5 and 50% isopropanol. Precipitated RNA was dissolved in dH$_2$O for UPLC/MS analysis.

To determine the amount of NAD that is covalently linked to RNA, the RNA samples were washed three times with 400 μL of 8.3 M urea, one time with dH$_2$O, two times with 4.15 M urea, and again four times with dH$_2$O in Amicon Ultra-0.5 mL Centrifugal Filter Units 10 kDa, to remove non-covalently bound cellular NAD. The recovered RNA was subsequently concentrated. The pulldown RNA samples or 10 μg urea-washed total RNA samples were treated with 10 μM NudC in the presence of 10 mM MgCl$_2$, 0.6 ng mL$^{-1}$ d4-riboside nicotinamide as IS, and 0.05 U μL$^{-1}$ alkaline phosphatase at 37 °C for 2 h. The same reaction with a catalytically inactive NudC mutant served as background reference. The reactions were filtered through Amicon Ultra-0.5 mL Centrifugal Filter Units 10 kDa to remove the enzymes and washed additionally four times with 200 μL dH$_2$O. The flow-through contained the cleaved nicotinamide riboside (NR), resulting from the NudC treatment of NAD-RNA and d4-NR as IS. It was collected and dried under vacuum. The amount of NR was determined by UPLC-MS/MS and reflects the exact same amount of original NAD-RNA in the digested sample. The employed UPLC-MS/MS setup contained a triple stage quadrupole mass spectrometer (Waters, Xevo TQ-S) system coupled with an Acquity UPLC system. A BEH Amide column (1.7 μm, 2.1 × 50 mm) was used with an eluent A (0.01% (v/v) aqueous formic acid with 0.05% ammonia and 5% acetonitrile) and B (acetonitrile included 0.01% formic acid), at a flow rate of 0.8 mL min$^{-1}$. The gradient started to change from 12.5% A/87.5% B to 95% A/5% B within 1.8 min. The ratio was changed back to starting conditions within the following 1.0 min. The column was had been pre-equilibrated under starting conditions for 1.0 min. Electrospray ionization was performed with a 1500 V capillary voltage, 11 V cone voltage, 150 °C

source temperature, 200 °C desolvation temperature, 150 L h$^{-1}$ cone nitrogen gas flow, and 800 L h$^{-1}$ desolvation gas flow (N$_2$). The Xevo TQ-S was automatically tuned to d4-NR and NR using the MassLynx V4.1 system software (Waters) with the IntelliStart standard procedures. Multiple reaction monitoring measurements were conducted, using collision gas (argon, 0.15 ml min$^{-1}$) for collision-induced decomposition and MS/MS transitions were monitored in the positive ion mode (N-ribosylnicotinamide: $m/z$ 254.94–122.81, d4-N-ribosylnicotinamide $m/z$ 258.94–126.81, 20 V, 50 mS dwell time for each mass transition).

**Gel electrophoresis.** Native PAGE (10%) was utilized to size select cDNA for NGS. Briefly, 10% acrylamide/Bis solution (19 : 1), 0.1% Ammonium persulfate(APS) (w/v), and 0.1% N,N,N′,N′-Tetramethylethylenediamine(TEMED) (v/v) along with 1× Tris-borate-EDTA (TBE) in 50 mL volume were mixed and poured between glass plates (19 cm × 27 cm). Gel mixtures were polymerized at room temperature for 45 min. The electrophoresis conducted at a stable 27 mA current for 2.5 h. The gel was then stained with SYBR Gold (Thermo Scientific) in 1× TBE buffer, 5 min. The signal intensities were read out by scanning the gel at 400 V, 50 or 100 μm resolution using a Typhoon FLA 9500. The printout picture of the gel (in its original size) was used for excision of desired size ranges within the corresponding gel lanes. APB gel electrophoresis was utilized to separate NAD-RNA, m$^7$G-RNA, and p/ppp-RNA from each other. Briefly, 0.5% (w/v) APB (Lab stock), 10% acrylamide/Bis solution (19:1), 0.1% (w/v) APS, and 0.1% (v/v) TEMED with 2.5× Tris-acetate-EDTA (TAE) buffer in 50 mL volume were mixed and poured between glass plates (Bio-Rad). Gels were run at a stable current (15–29 mA per gel) in 1× TAE buffer. Gels were then stained with SYBR Gold or gels, containing $^{32}$P-labeled nucleic acids, were exposed to storage phosphor screens (GE Healthcare) and visualized using a Typhoon FLA 9500. Signal quantification was performed using the ImageQuant software (GE Healthcare).

**In vitro transcription variants.** Radio-labeled RNA: 400 nM double-stranded DNA (dsDNA) served as a template in a transcription buffer containing 40 mM Tris-HCl pH 7.9, 1 mM spermidine, 22 mM MgCl$_2$, 0.01% Triton X-100, 10 mM dithiothreitol (DTT), 5% dimethyl sulfoxide, along with 60 μCi α-$^{32}$P-ATP, 10 μCi α-$^{32}$P-CTP, 4 mM each CTP/GTP/UTP, 2 mM ATP, 6 mM NAD (NAD-RNA) or without NAD (ppp-RNA), 0.7 μM T7 polymerase (self-prepared) within a total reaction volume of 100 μL. The in vitro transcription reaction was incubated at 37 °C for 3 h. Subsequently, 10 U DNase I were added and the reaction incubated for an additional 30 min at 37 °C. ppp-RNA was purified by 10% denaturing PAGE, as described above. NAD-RNA was first purified by 10% denaturing PAGE, followed by a 10% PAGE, supplemented with 0.5% APB, to remove ppp-RNA. For p-RNA preparation, 1 μg ppp-RNA was treated with 20 U RNA 5′-polyphosphatase and 1× Reaction Buffer (Epicentre) in a reaction volume of 20 μL for 1 h at 37 °C. The desired p-RNA species was then obtained by performing P/C/I extraction twice, followed by ethanol precipitation, as described. To generate m$^7$G-RNA, ~1.2 μM denatured ppp-RNA (65 °C, 5 min) was first added to a mixture of 1× Scriptcap capping buffer, 1 mM GTP, fresh 0.1 mM S-adenosyl methionine (SAM), and 1 U μL$^{-1}$ Script Guard RNase Inhibitor. The mixture was then supplemented with 0.4 U μL$^{-1}$ Scriptcap Capping Enzyme, in a final reaction volume of 50 μL and incubated for 50 min at 37 °C. The modified RNA reaction was then purified by 10% APB-PAGE as described above, to separate uncapped, ppp-RNA, and m$^7$G-capped RNA.

Luciferase mRNA: 140 nM (~6 μg) linear dsDNA template was added into the same transcription buffer as described above, along with 4 mM each CTP/GTP/UTP, 2 mM ATP, 6 mM NAD (NAD-mRNA) or without NAD (ppp-mRNA), 0.7 μM T7 polymerase (self-prepared) with a total reaction volume of 100 μL. The reaction mixture was incubated for 3 h at 37 °C. Similarly, 10 U DNase I were added subsequently to the reaction and incubated for an additional 30 min at 37 °C. RNA integrity was examined by visualization of the reaction products on a 1% formaldehyde-denaturing agarose gel. RNA was purified by performing P/C/I extraction twice and precipitated with ethanol. Precipitated RNA was loaded onto Amicon Ultra-0.5 mL Centrifugal Filter Units 10 kDa and washed four times with 400 μL dH$_2$O to remove free NTPs and small molecules. Retained RNA was collected and again precipitated with ethanol. Recovered RNA was then washed twice with 75% ethanol and ultimately dissolved in 20 μL dH$_2$O. For further isolation of 5′-NAD-modified RNA, 5 μg partially purified NAD-RNA was treated with 5 U RNA 5′-polyphosphatase in 1× Reaction Buffer (Epicentre) with an overall volume of 10 μL for 40 min at 37 °C. Remaining NAD-capped RNA was then P/C/I extracted twice and ethanol precipitated. Precipitated mRNAs were further treated with 2 U Xrn-1 in 1× NEBuffer 3 (NEB) reaching a total reaction volume of 40 μL. For p-mRNA preparation, 5 μg ppp-mRNA was treated with 10 U RNA 5′-polyphosphatase in 1× Reaction Buffer (Epicentre) and final volume of 20 μL for 40 min at 37 °C. For m$^7$G-mRNA preparation, 1 μM denatured ppp-mRNA (65 °C, 5 min) was mixed with 1× capping buffer (NEB), 0.5 mM GTP, fresh 0.1 mM SAM, and 0.5 U μL$^{-1}$ Vaccinia Capping Enzyme in 20 μL volume, incubate at 37 °C, 40 min. Processed RNA was purified twice by P/C/I extraction and precipitated with ethanol. All mRNA raw concentrations were measured by nanodrop, followed by qRT-PCR to determine the relative abundance of the mRNA middle region, as well as of the 3′-end region. Final concentration of mRNAs was adjusted accordingly.

**Fluorescence microscopy and colony fluorescence imaging.** For fluorescent live cell imaging, a DeltaVision Elite widefield fluorescence microscope (GE Healthcare) consisting of an inverted epifluorescence microscope (IX71; Olympus) equipped with a light-emitting diode light engine (seven-color InsightSSI module; GE Healthcare), an sCMOS camera (pco.edge 4.2; PCO), and a ×60 1.42 NA Plan Apochromat N oil-immersion objective (Olympus) was used. Cells were inoculated in low-fluorescence medium (SC medium prepared with yeast nitrogen base lacking folic acid and riboflavin; CYN6501, ForMedium) and grown to mid-log phase. Cells were immobilized for imaging in glass-bottomed 96-well plates (MGB096- 1-2-LG-L; Matrical) using Concanavalin A and fluorescence filters for GFP, as standard procedure[50]. For imaging the fluorescence of cell colonies and cell patches on plates, a custom-made fluorescence illumination cabinet was used, equipped with filters for GFP.

**Cell-free extract preparation[50] and in vitro translation.** Preparation of cell-free extracts for in vitro translation was performed as described[51] with minor modifications. Cell pellets were obtained from 0.5 L YPD yeast (BY4742 strain) culture (OD$_{600}$ ≈ 15). The cell pellet was washed three times with 50 mL ice-cold Breaking Basic Buffer (30 mM HEPES-KOH pH 7.6, 100 mM KOAc pH 7.0, 3 mM Mg (OAc)$_2$ pH 7.0, 2 mM DTT), supplemented with 8.5% (w/v) mannitol. Then, the wet weight of the cells was determined. The cells were subsequently resuspended in Breaking Basic Buffer with 8.5% mannitol and 0.5 mM phenylmethylsulfonyl fluoride (PMSF) (Sigma-Aldrich, dissolved in isopropanol), under addition of 1.5 mL per 1 g wet weight. The solution was supplemented with ice-cold, sterile ~0.5 mm glass beads, whereby the equivalent of six times of the cell wet weight was added. The cells were raptured by manual shaking (70 cm hand path, 2–5 Hz), in overall five rounds, each consisting of 1 min shaking interrupted by 1 min cooling on ice. Glass beads were removed by low speed centrifugation (1000 × g, 2 min, 4 °C), then the cleared supernatant was obtained by two consecutive rounds of centrifugation (29,000 × g, 20 min, 4 °C). The solution was filtered through a 0.45 μm filter and then 2 mL of the supernatant were subjected to Fast Protein Liquid Chromatography (FPLC) runs, outlined below. HiPrep 26/10 Desalting Columns (containing Sephadex G-25 Fine resin) were pre-equilibrated with 100 mL Breaking Basic Buffer, supplemented with 0.5 mM PMSF. The injected sample was resolved by the indicated column matrix running on a FPLC system (flow rate 1 mL min$^{-1}$, 0.5 mL collected fractions) employing the same equilibration buffer. Absorption values of each fraction were determined at 260 nm (A$_{260}$) and appropriate fractions, exceeding 75% of the highest A$_{260}$ value, were pooled together. Next, 1 mM CaCl$_2$ and 50 U mL$^{-1}$ micrococcal nuclease were added to the pooled fractions, followed by incubation at 26 °C for 15 min. The reaction was stopped by adding EGTA to a final concentration 2.5 mM. Aliquots of 100 μL each were then flash-frozen in liquid nitrogen and stored at −80 °C.

In vitro translation: Master Translation Solution was freshly prepared by mixing 25 mM HEPES-KOH pH 7.6, 1.25 mM ATP, 0.125 mM GTP, 0.15 U μL$^{-1}$ creatine phosphokinase, 2.5 mM DTT, 125 mM KOAc, 5 mM MgOAc pH 7.0, 25 μM Amino Acid Mixtures, 1 U μL$^{-1}$ RNas in Ribonuclease Inhibitors, 80 nM m$^7$G-capped *Renilla* mRNA, and dH$_2$O in a final volume 80 μL. The solution was gently mixed and 4 μL of the Master Translation Solution aliquoted to individual PCR tubes. Then, 1 μL mRNA (200 ng or accordingly) and 5 μL of the cell-free extract were added to thus prepared reaction volumes. The "ready-for-translation" solution was again mixed gently and incubated at 26 °C for 30 min. Next, 90 μL dH$_2$O were added to each in vitro translation reaction. Seventy five microliters of this dilution were then used to conduct a firefly luciferase activity assay (Bright-Glo Luciferase Assay System). The remaining 25 μL, with additional 50 μL of dH$_2$O, were subjected to a *Renilla* luciferase activity assay (*Renilla*-Glo Luciferase Assay System), executed according to the manufacturing instructions. Emitted luminescence was read out using a TECAN plate reader.

**Cellular NAD quantification.** The general experimental procedure was based on the manufacturer's protocol (NAD/NADH Quantification Kit) with minor modification. Yeast cells were cultured in 50 mL YPD medium in biological quadruplicates ($n = 4$). One milliliter of these cultures, at an OD$_{600}$ of ~0.8, were pelleted by the centrifugation at 4000 × g, 1 min, 4 °C. The cell pellets were washed four times with 1 mL ice-cold PBS and resuspended in 1 mL NADH/NAD Extraction Buffer. The resuspension (400 μL) were subjected to three freeze–thaw cycles, each consisting of 10 min on dry ice, alternating with 10 min thawing at room temperature. The homogenized cell lysates were vortexed for 10 s and centrifuged at 14,000 × g, 4 °C, 10 min. Twenty microliters of each supernatant were aliquoted for subsequent lysate RNA quantification. The remaining solution was applied onto a 10 kDa spin filter to remove proteins larger than 10 kDa, by centrifugation at 14,000 × g, 4 °C, 10 min. The flow-throughs were collected and NAD/NADH quantification was conducted, as stated by the manufacturer. The determined NAD amounts were normalized by the amount of measured lysate RNA.

**RNA in vitro decapping and NAD hydrolysis kinetic assays.** In general, $^{32}$P-body-labeled NAD-/m$^7$G-capped RNAs or $^{32}$P-labeled NAD were incubated with 0.4–1.6 μM recombinant proteins in 40 μL decapping reaction containing 25 mM Tris-HCl pH 7.5, 50 mM NaCl, 50 mM KCl, 2 mM MgCl$_2$, 1 mM DTT, 1 mM MnCl$_2$ (was absent in reactions referred to as without Mn$^{2+}$ kinetics) and

incubated at 37 °C for 120 min. Samples that were treated with the *E. coli* Nudix hydrolase NudC are referred to as "positive control." Aliquots of 5 μL were taken from the reaction mixtures at indicated time points. Thus, treated RNAs were mixed with the same volume of 2× APB Gel Loading Buffer (50 mM Tris-HCl pH 7.5, 8 M Urea, 20 mM EDTA, 20% Glycerol, 0.01% Xylene Cyanol, 0.01% Bromophenol Blue) and placed on ice for further APB gel electrophoresis. Reaction mixtures, containing NAD, were stored on ice before performing thin-layer chromatography (TLC; DC-Fertigfolie ALUGRAM Xtra SIL G/UV254, 20 cm × 20 cm). Resolution of nucleotides via TLC, at room temperature for 5.5 h, was achieved employing a flow phase of 1 M NH4OAc and ethanol (4 : 6).

**Determination of RNA NAD-capping ratios**. For each assay, 100 μg total RNA (gDNA free) was subjected to ADPRC treatment as fully treated group. An equal amount of total RNA was applied to the same treatment without ADPRC as background group. The subsequent copper-click reaction, capture by streptavidin beads (streptavidin-unbound flow-through RNAs (non-NAD-RNA) were collected and precipitated with ethanol), as in the standard NAD captureSeq procedure[11]. For RT on beads, to each sample of the fully treated group, as well as the background group, were added 2.5 μM random hexamers, 0.5 mM dNTP mix, and dH2O at 65 °C for 5 min. After reaction, the beads were incubated on ice for 2 min. Then, to the reaction was added 1× SSIV Buffer (Thermo Scientific), 5 mM DTT, 10 U μL$^{-1}$ SuperScript IV Reverse Transcriptase, and 50 ng μL$^{-1}$ acetylated BSA (Sigma-Aldrich), and the mixture incubated at 23 °C for 10 min, followed by 1 h at 55 °C. This was followed by RNA rebinding to streptavidin beads, washing, NaOH-mediated hydrolysis, and cDNA precipitation, as described in the NAD captureSeq protocol[11]. Equivalents of 1 μg of diluted non-NAD-RNAs were reverse transcribed, employing random hexamer oligos, as described above. The relative abundance of transcripts was then quantified by RT-qPCR. The enrichment of transcripts, assessed by RT-qPCR of cDNA, from the fully treated group, the background group, and the non-NAD-cap group was normalized by rRNA, RDN5-1. The NAD-modification ratio of each RNA species was determined by the equation NAD-modification ratio = (NAD-RNA$_{\text{fully treated group}}$ − unspecific-binding-RNA$_{\text{background group}}$)/(NAD-RNA$_{\text{fully treated group}}$ + other-cap-RNA$_{\text{non-NAD-cap group}}$).

For the NAD-TDH3 promoter assay, 100 μg total RNA (gDNA free) were supplemented with 5 ng of NAD-RNAIII (self-prepared) and 5 ng ppp-RNAI (self-prepared), and subjected to the same ADPRC treatment and copper-click reaction, as described above. Then, 50 μL of Hydrophilic Streptavidin Magnetic Beads slurry per reaction was utilized to enrich for NAD-modified RNAs in a 96-well plate (flat-bottom, Corning) format. The beads were hereby pre-washed twice with 150 μL Immobilization Buffer (10 mM HEPES pH 7.2, 1 M NaCl, 5 mM EDTA) and then blocked using acetylated BSA (100 μg mL$^{-1}$, Sigma) in 100 μL Immobilization buffer at room temperature for 30 min under gentle agitation. Subsequently, beads were washed three times with 150 μL Immobilization Buffer. Precipitated RNA from the copper-click reaction was dissolved in 100 μL Immobilization Buffer and then added onto the washed beads. The mixture was incubated at room temperature, while shaking at 500 r.p.m. for 1 h. Next, the 96-well plate was placed on a magnetic rack at room temperature for 10 min. Then, the first supernatant was collected. Beads were washed with 100 μL Immobilization Buffer and following that, the second supernatant was collected and pooled with the first one, subsequently referred to as "non-NAD-RNAs." The RNA from the pooled supernatants was precipitated and 1 μg of diluted non-NAD-RNAs was utilized for RT to determine transcript abundance by RT-qPCR. RNAs captured by magnetic streptavidin beads (+ADPRC for S group, −ADPRC for N group) were washed five times with 150 μL Streptavidin Wash Buffer (8 M Urea, 10 mM Tris-HCl pH 7.4) and then washed three times with 150 μL First strand Buffer (25 mM Tris-HCl pH 8.3, 37.5 mM KCl, 1.5 mM MgCl2). Subsequently, the beads were re-blocked using 100 μL First strand Buffer, containing acetylated BSA (100 μg mL$^{-1}$). Then, equilibrated beads were washed twice with 150 μL 1× SSIV buffer (Thermo Scientific). RT was conducted using random hexamer oligos. The reactions were then transferred into PCR tubes and heated to 65 °C for 5 min to denature the RNA. The tubes were cooled on ice for 5 min, then incubated at 23 °C for 10 min, followed by a heating step at 55 °C for 30 min to enable the RT reaction by Superscript reverse transcriptase IV. Next, 100 μL NaOH (0.15 M) were added into the reaction and the mixture incubated at 55 °C for 15 min. The first RT supernatant was collected. The remaining beads were repeatedly subjected to the same treatment, by adding 100 μL NaOH (0.15 M). Then the second, residual RT supernatant was pooled with the first one and is subsequently referred to as NAD-RNA (S group) and background RNA (N group). Both were ethanol precipitated and directly subjected to RT-qPCR. The abundances of NAD-TDH3 RNA and non-NAD-TDH3 RNA were normalized by transcript abundances of the NAD-RNAIII and ppp-RNAI controls, respectively. The NAD-capping ratio of TDH3 was calculated by dividing NAD-TDH3 by the sum of NAD-TDH3 and non-NAD-TDH3.

For in vitro-transcribed NAD/ppp-mRNAs, 200 ng of denatured (~375 fmol) NAD-mRNAs were supplemented with 10 mM Tris-HCl pH 8.0, 50 mM NaCl, 2 mM fresh DTT, and 1 μM DNAzyme (design procedure as described[52], sequences are listed in Supplementary Table 2) in a total volume 9 μL at 65 °C, 5 min. The reaction was cooled to 37 °C. The reactions were then supplemented with 1 μL 500 μM MgCl2 and incubated for an additional 1 h at 37 °C. Reactions containing ppp-mRNAs are referred to as "positive control," whereas reactions containing NAD-

mRNA in the absence of DNAzyme are referred to as negative control. Reactions containing NAD-mRNA and an additional 2 μM NudC (added after denaturation) served as reference RNA. The reactions were stopped by adding an equal volume of 2× APB Gel Loading Buffer. Separated and cleaved products of NAD-RNA and ppp-/p-RNA were visualized by APB-supplemented PA gel electrophoresis and subsequent SYBR Gold staining. The gel was scanned using a Typhoon imager for further signal quantification. For those samples with weak signal intensities upon DNAzyme treatment on APB gel, RT-qPCR was performed to quantify RNA abundance.

**RT-qPCR and standard PCR procedures**. Real-time PCR was performed using 250 nM Fw/Rev primer, 5 μL diluted cDNA, and 1× SsoAdvanced Universal SYBR Green Supermix in 20 μL reaction. PCR conditions were the following, denaturing step at 95 °C (2 min), 40 cycles of consecutive annealing/extension steps at 95 °C for 7 s and 60 °C for 15 s, respectively. Melting curves were generated by heating from 65 °C to 95 °C with an incremental increase of 0.5 °C s$^{-1}$. Fluorescence was measured throughout using a LifeCycle 480 Instrument.

Two rounds of PCR were carried out to generate linear DNA templates for in vitro translation reactions as follows: the first round aimed to bridge the 5′-UTR/CDS to the firefly Luc2 sequence. The reactions contained 6.4 pM 5′-UTR/CDS template, 6.4 pM bridge region DNA (Supplementary Table 2), and 6.4 pM firefly Luc2 DNA template, 200 nM dNTPs, 1× Q5 Reaction Buffer (NEB), 0.02 U μL$^{-1}$ hot-start Q5 high-fidelity DNA polymerase in a total volume 50 μL. The PCR was initiated by heating to 98 °C for 40 s and followed by 5 cycles (98 °C 10 s, 65 °C 20 s, 72 °C 2 min). The second round of the PCR aimed to specifically amplify the bridged 5′-UTR/CDS, bearing the Luc2 template extension, by additionally adding corresponding 500 nM Fw_5UTR primer, 500 nM Rev_Luc2_polyA primer, and 1× Q5 Reaction Buffer (NEB) to the final volume of 56.2 μL.

All other PCRs were performed using the Q5 high-fidelity DNA polymerase for amplifying NAD captureSeq cDNA with barcodes, as described[11] and related dsDNA templates for in vitro translation assay, as linear templates from plasmids. Otherwise, *Taq* polymerase/Q5 high-fidelity DNA polymerase were employed to amplify 5′-UTR/CDS sequences from gDNA and from plasmids, obtained by using standard PCR procedures.

**Quantification**. PAGE gel, APB gel, and TLC intensities were quantified using the ImageQuant software (GE Healthcare).

**Statistical analysis**. All samples for NAD captureSeq, transcriptomics, UPLC-MS, RT-qPCR, and proteomics were prepared as biological triplicates. The outlined in vitro experiments, including NAD kinetics, RNA decapping kinetics, and in vitro translations, were performed as technical triplicates. Mean values ($n \geq 3$) with SDs, and Student's *t*-test (single-tail and double tail (only FCM data), unequal variance) was calculated by R/python. For NGS TSS switching relevant math, covariance with two variables, linear trend test, and corresponding *p*-values, as well as FDR value, were calculated using python scripts. For NGS NAD captureSeq, NAD-RNA enrichment and transcriptome differential expression analysis, DESeq2 was utilized, as described above.

**Reporting summary**. Further information on research design is available in the Nature Research Reporting Summary linked to this article.

## Data availability
NGS raw data and analyzed files are available in the GEO repository under the GEO Accession: GSE146368. Proteomics raw data and analyzed files are deposited in the ProteomeXchange Consortium via the PRIDE repository: PXD017893. The data supporting the findings of this study are available from the corresponding authors upon reasonable request. Source data are provided with this paper.

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

## Acknowledgements
We thank Jäschke Lab members, V. Winkler (Heidelberg University), and A. Hotz-Wagenblatt (DKFZ, Heidelberg) for discussions; H.C. Lee and D. Grimm for strains and plasmids; M. Brunner for access to LightCycler, ZMBH Flow Cytometry & FACS Core Facility for the FCM measurement; M. Rettel and F. Stein (EMBL Proteomics Core Facility, Heidelberg) for proteomics analysis; and bwHPC (bwForCluster MLS&WISO and bwUniCluster) for cluster computation resources. This work was supported by the German Research Foundation (DFG, grant number Ja794/10, SPP 1784, to A.J.).

## Author contributions
Conceptualization: Y.Z. and A.J. Methodology: Y.Z., D. Kuster, T.S., D. Kirrmaier, G.N., D.I., and V.B. Investigation: Y.Z., D. Kuster, T.S., D. Kirrmaier, G.N., D.I., and H.H. Formal analysis: Y.Z., M.K., H.H., and A.J. Supervision: H.H., M.K., and A.J. Administration: A.J. Writing—original draft: Y.Z. and A.J. Writing—review and editing: all authors.

## Funding

## Competing interests
The authors declare no competing interests.
