## [Peer Review File · Nature Communications]

REVIEWER COMMENTS

Reviewer #1 (Remarks to the Author):

The manuscript from Zhang et al. provides important new insight into the biological significance of RNA 5'-end capping with the nucleoside-containing metabolite NAD. The Jashke lab has been at the forefront on studies of this atypical 5'-end modification beginning with their seminal 2014 Nature paper reporting the development of "NAD captureSeq" as a method to identify NAD-capped RNA in the transcriptome of any organism. In this work, they apply NAD captureSeq to analyze the extent of NAD capping in yeast.

Although a prior study identified a handful of NAD-capped transcripts in yeast (Walters et al. PNAS, 2017), this prior study used a NAD captureSeq procedure that excluded small RNAs. Thus, the Jashke lab has repeated the NAD captureSeq analysis in yeast using procedures enabling analysis of both full-length and small RNA products. The results reveal an abundance of NAD-capped short RNA products that were missed in the prior work. By itself, this is an important contribution to the field. The authors further report effects of the NAD-decapping enzymes, Rai1, Dxo1 and Npy1, on the distributions of NAD capped species, and show that NAD-capped products are not translated in yeast (at least in the cytoplasm). The work is worthy of publication in Nature Communication. I do have a few modifications to the text that I strongly recommend. These changes would restate some of the conclusions the authors make that I do not believe are warranted based on the data provided.

(1) The authors state that nearly all of the detected RNA products are generated by use of NAD by Pol II as an initiating substrate. While I believe this to be true, the authors did not provide a sufficient justification to support this conclusion.

(2) The authors state that their analysis of rai1, dxo1 and npy1 knockouts support a role for these enzymes in processing NAD caps in vivo. While this probably is true, the data do not support the conclusion. Rather, the results of the knockouts are quite confusing and suggest the processing of NAD caps and the role of these enzymes in the processing may be quite complex. For example, the simple prediction of knocking out one or more of these enzymes would be that one would expect more NAD-capped products. But this is not observed, rather it appears in some cases that the abundance of NAD-capped RNAs actually decreases. This is further complicated by the reported change in overall abundance of a significant number of transcripts in the npy1 deletion mutant. I would strongly suggest the text is rewritten to more accurately reflect that the analysis of the rai1, dxo1 and npy1 knockouts may be consistent with a role of these factors in processing NAD capping, but by no means demonstrative evidence. The analysis of both NAD and ATP levels (which will compete with NAD during initiation) in the knockouts may be informative in sorting things out.

(3) The authors description of the significance of the YAAG motif in NAD-capping should be rewritten. The text makes misleading comparisons between "consensus" sequences for NAD capping. In particular, the comparisons between the YAAG sequence and previously reported consensus sequences for E. coli RNAP and mtRNAP are misleading. This is because E. coli RNAP and mtRNAP consensus sequences were

determined by comparing the efficiency of initiation with NAD vs the efficiency of initiation with ATP. Thus, for each promoter sequence, yields of NAD-capped RNA were compared with yields of total RNA (NAD capped + uncapped RNA) to determine how promoter sequence modulates yields of NAD-capped RNA relative to uncapped RNA. In contrast, the YAAG sequence reported in this work, was determined by comparing yields of NAD-capped products for different promoters, without considering other RNA products produced from these promoters. Thus, comparing the YAAG consensus with the E. coli RNAP or mtRNAP consensus is comparing apples to oranges. To avoid confusion, the authors should rewrite the text to explain how each consensus sequences was derived, making it clear that they should not be directly compared with one another.

Minor points.

(4) Labels on Figures 5A-B, 6A-B, and extended data Figure 5 are confusing. One letter codes for each knockout strain should be changed for clarity. Where appropriate, colors used in graphs should be explained in the figure legends.

Reviewer #2 (Remarks to the Author):

The 5'-cap structure is a characteristic feature of eukaryotic mRNAs and lncRNAs, and plays a critical role in RNA stabilization, splicing, nuclear export and translation. Other than m7G cap, NAD-cap is a new type of cap structure found in bacteria and eukaryotes. The research group lead by Andrea Jaschke made a pioneer work identifying NAD-capped RNA in bacteria with chemo-enzymatic method assisted by deep seq called NAD captureSeq.

According to the previous study in budding yeast, small numbers of mRNAs are NAD-capped, questioning the importance of NAD capping in yeast. This manuscript aims to map NAD-capped RNA in *Saccharomyces cerevisiae* with their original method. They detected >1400 species of NAD-RNAs, most of which are short less than 170 nt and low in frequency (>5%). RNA polymerase II accidentally uses NAD as a primer for transcription start. They identified a YAAG core motif in the promoter for efficient incorporation of NAD in the nascent chain. NAD-capped RNA is not capable for translation in vitro, indicating that NAD-cap seems to be disadvantageous to the cell. The experiments were well executed, but data presentation and description should be improved for the revised version.

Regarding RNA length and NAD capping, why was >170 nt chosen? The authors need to show and compare size distribution of transcripts with and without NAD cap. If NAD capping occurs in short RNA, Northwestern blotting with Streptavidin HRP of the biotinylated total RNA will provide direct evidence for that.

In the abstract, it says >5% frequency of NAD capping. But, this reviewer cannot find any information on this figure in the manuscript.

They performed the NAD capture with total RNA isolated from WT and several KO strains. Quality of total RNA might be different in such mutant cell lines. They need to show pictures of electrophoresis or bioanalyzer data of each total RNA to confirm its quality.

Regarding Npy1 decapping, NAD capping was upregulated in only two transcripts (Fig. 2h), although NAD amount of total RNA decreased (Fig. 2f). The authors need to explain for this discrepancy.

LC/MS data for NAD cap should be shown in supplementary figure.

mRNAs of LSM7 and RSM10 seem to be good substrates for Npy1 in the cell. The authors should analyze decapping efficiencies of these transcripts by recombinant Npy1 in vitro, and compare the activity with TDH3 mRNA.

Other comments

Poor information in figure legends.

For example, no information for bar graphs in Fig. 1ac.

Pie chart in Fig. 1b. Is this a fraction of gene number or read number?

Reviewer #3 (Remarks to the Author):

Review for the manuscript NCOMMS-20-22502-T by Dr. Jäschke and co-authors entitled "Extensive 5'-Surveillance Guards Against Non-Canonical NAD-Caps of Nuclear mRNAs in Yeast."

The ubiquitous redox coenzyme nicotinamide adenine dinucleotide (NAD) acts as a non-canonical cap structure on prokaryotic and eukaryotic ribonucleic acids. NAD-RNAs were found in human cells, and the NAD cap promotes RNA decay through DXO-mediated removal of 5' NAD. In yeast, only 3' RNA species and low abundance were reported.

In this study, the whole landscape of NAD transcripts in yeast was demonstrated using the original NAD capture-Seq protocol. NAD is incorporated with low efficiency (<5%) during the initiation step by RNA polymerase II, which uses distinct promoters with a YAAG core motif. NAD-RNAs mostly correspond to mRNA 5'-ends and NAD mRNAs are 3'-truncated and poorly translated. The authors discovered the Nudix pyrophosphohydrolase 1 (Npy1) as a new member of guard for NAD-RNA in yeast and performed a comprehensive analysis of three NAD-RNA decapping factors, Rai1, Dxo1, and Npy1. Three decapping enzymes guard against NAD-RNA at different cellular locations, targeting overlapping transcript

populations. The authors propose that NAD incorporation into RNA seems to be disadvantageous to the cell, and different surveillance machinery is responsible for decapping and rejection of NAD-RNAs. The proposal is potentially interesting. However, the physiological relevance of the surveillance machinery for NAD-RNA is unclear. There is no direct evidence demonstrating that NAD incorporation into RNA seems to be disadvantageous to the cell. It is also unclear whether the defects in the clearance of NAD-RNAs directly cause RNA changes in the mutants. The growth defects RAI1 deletion may be due to the defects in removing the entire cap structure dinucleotide from an mRNA. Also, the deletion of the other two factors did not confer the significant defects in the growth. Therefore, there is no significant suggestion for the function of the NAD cap in yeast.

1) It was reported that Rai1 functions to clear mRNAs in cells subjected to glucose starvation or amino acid starvation by preferential hydrolysis of unmethylated capped RNA. To clarify the in vivo function of the scavenger system for the removal of NAD at the 5' end of mRNA, the authors should clarify the defects of the NAD-RNA's surveillance machinery in translation or RNA homeostasis. Since it is demonstrated that NAD mRNAs are poorly translated, it should be examined whether NAD-RNAs (TDH3, etc.) are rapidly degraded in cells subjected to stress conditions, including glucose starvation or amino acid starvation.

2) Given that mitochondria mRNAs were highly 5'-NAD modified, the authors propose a regulatory role of the NAD cap in mitochondria. Please discuss the possible function of three factors in the decapping of mitochondrial NAD-RNA.

3) It was reported that DXO/Rai1 enzymes remove 5'-end FAD and dephospho-CoA caps on RNAs. Is it possible to discuss the possibility that Npy1 to remove the modifications at 5'-end of RNAs?

4) Please discuss the putative mechanism for RNAP II to select a different TSS for initiating transcription with NAD, and possible mechanism to recruit Rai1 to the RNAP elongation complex that initiates from TSS.

Manuscript NCOMMS-20-22502-T

Point-by-point response to the reviewers' comments

We kindly ask to review our changes in the author-supplied Word (.docx) manuscript file, as the Nature website's PDF converter removes all markups (including "Track changes") and alters line numbers.

Reviewer #1 (Remarks to the Author):

The manuscript from Zhang et al. provides important new insight into the biological significance of RNA 5'-end capping with the nucleoside-containing metabolite NAD. The Jashke lab has been at the forefront on studies of this atypical 5'-end modification beginning with their seminal 2014 Nature paper reporting the development of "NAD captureSeq" as a method to identify NAD-capped RNA in the transcriptome of any organism. In this work, they apply NAD captureSeq to analyze the extent of NAD capping in yeast.

Although a prior study identified a handful of NAD-capped transcripts in yeast (Walters et al. PNAS, 2017), this prior study used a NAD captureSeq procedure that excluded small RNAs. Thus, the Jashke lab has repeated the NAD captureSeq analysis in yeast using procedures enabling analysis of both full-length and small RNA products. The results reveal an abundance of NAD-capped short RNA products that were missed in the prior work. By itself, this is an important contribution to the field. The authors further report effects of the NAD-decapping enzymes, Rai1, Dxo1 and Npy1, on the distributions of NAD capped species, and show that NAD-capped products are not translated in yeast (at least in the cytoplasm). The work is worthy of publication in Nature Communication. I do have a few modifications to the text that I strongly recommend. These changes would restate some of the conclusions the authors make that I do not believe are warranted based on the data provided.

We thank the reviewer for the positive assessment of our work, and for stating that it is an important contribution to the field, worthy of publication in Nature Communications.

(1) The authors state that nearly all of the detected RNA products are generated by use of NAD by Pol II as an initiating substrate. While I believe this to be true, the authors did not provide a sufficient justification to support this conclusion.

The reviewer is right. We toned down our claim and rephrased the text. In particular:

- The 3rd sentence of the abstract (p. 2, l. 22) now reads: "NAD incorporation occurs mainly during transcription initiation by RNA polymerase II, ..."

- On page 9, l. 217 we changed the sentence into: "This analysis implies that our identification procedure revealed real TSSs and further supports the assumption that transcriptional NAD incorporation is the predominant biosynthetic pathway to NAD-RNAs."

- on page 15, l. 357 we added the sentence: "This finding does not rule out the existence of unknown alternative post-transcriptional pathways for NAD incorporation, e.g. for enriched snoRNAs or rRNA, or for mRNAs without the YAAG motif."

(2) The authors state that their analysis of *rai1*, *dxo1* and *np1* knockouts support a role for these enzymes in processing NAD caps *in vivo*. While this probably is true, the data do not support the conclusion. Rather, the results of the knockouts are quite confusing and suggest the processing of NAD caps and the role of these enzymes in the processing may be quite complex. For example, the simple prediction of knocking out one or more of these enzymes would be that one would expect more NAD-capped products. But this is not observed, rather it appears in some cases that the abundance of NAD-capped RNAs actually decreases. This is further complicated by the reported change in overall abundance of a significant number of transcripts in the *np1* deletion mutant. I would strongly suggest the text is rewritten to more accurately reflect that the analysis of the *rai1*, *dxo1* and *np1* knockouts may be consistent with a role of these factors in processing NAD capping, but by no means demonstrative evidence. The analysis of both NAD and ATP levels (which will compete with NAD during initiation) in the knockouts may be informative in sorting things out.

True. We fully agree. In the introduction (p. 3, l. 48) we added the word "likely".

In the *Npy1* chapter, we changed last sentence (p.7, l. 155) to read: "Collectively, these data are consistent with a role of *Npy1* in processing NAD caps *in vivo*."

On p.8, l. 179, we changed the concluding sentence by borrowing the sentence suggested by this reviewer: "Thus, our analysis of the *rai1Δ*, *dxo1Δ* and *np1Δ* knockout strains may be consistent with

a role of the affected gene products in processing NAD capping, but by no means demonstrative evidence.”

We thank the reviewer for suggesting analysis of NAD and ATP levels. A new PhD student, scheduled to start in October, will investigate the influence of numerous variables, including NAD and NTP concentrations, on NAD capping in yeast and in her first year establish the analytical methods and protocols.

(3) The authors description of the significance of the YAAG motif in NAD-capping should be rewritten. The text makes misleading comparisons between “consensus” sequences for NAD capping. In particular, the comparisons between the YAAG sequence and previously reported consensus sequences for *E. coli* RNAP and mtRNAP are misleading. This is because *E. coli* RNAP and mtRNAP consensus sequences were determined by comparing the efficiency of initiation with NAD vs the efficiency of initiation with ATP. Thus, for each promoter sequence, yields of NAD-capped RNA were compared with yields of total RNA (NAD capped + uncapped RNA) to determine how promoter sequence modulates yields of NAD-capped RNA relative to uncapped RNA. In contrast, the YAAG sequence reported in this work, was determined by comparing yields of NAD-capped products for different promoters, without considering other RNA products produced from these promoters. Thus, comparing the YAAG consensus with the *E. coli* RNAP or mtRNAP consensus is comparing apples to oranges. To avoid confusion, the authors should rewrite the text to explain how each consensus sequences was derived, making it clear that they should not be directly compared with one another.

Good point. We agree. On page 15, l. 361 we added the sentence: “It should, however, be noted that the *E. coli* RNAP and yeast mtRNAP consensus motifs were established using an entirely different methodology, making a direct comparison difficult.”

Minor points.

(4) Labels on Figures 5A-B, 6A-B, and extended data Figure 5 are confusing. One letter codes for each knockout strain should be changed for clarity. Where appropriate, colors used in graphs should be explained in the figure legends.

Following the reviewer’s suggestions, we made the following changes to those figures:

Fig. 5a-b (new number Fig. 7a-b): we changed the figure title “drn(S/N)_sharpA_top25_motif10” into full name “*dxo1Δ rai1Δ npy1Δ*(S/N)_sharpA_top25_motif10”, while “drn(S/N)_sharpA_Background25_motif10” into “*dxo1Δ rai1Δ npy1Δ*(S/N)_sharpA_Background25_motif10”. In the figure caption (page 29, l.585) we added the statement “The letters, representing the four nucleobases, were colored dark orange for A/T, and blue for G/C.” into the figure legend.

Supplementary Fig. 5 a-m and Fig. 6a-b (new number Fig. 8a-b): we changed all the abbreviation names of knockout strains into the respective full names.

For consistency, we also adopted this change to the strain abbreviations throughout the manuscript, in particular in Fig. 3a,b,d (new number Fig. 5a,b,d), Extended Fig. 3a-e, Fig. 4b (new number Fig. 6b), Supplementary Fig. 4d, Fig. 5a-b (new number Fig. 7a-b), Fig. 6f (new number Fig. 9b), and Supplementary Fig 6a-b.

Reviewer #2 (Remarks to the Author):

The 5'-cap structure is a characteristic feature of eukaryotic mRNAs and lncRNAs, and plays a critical role in RNA stabilization, splicing, nuclear export and translation. Other than m7G cap, NAD-cap is a new type of cap structure found in bacteria and eukaryotes. The research group lead by Andrea Jaschke made a pioneer work identifying NAD-capped RNA in bacteria with chemo-enzymatic method assisted by deep seq called NAD captureSeq.

According to the previous study in budding yeast, small numbers of mRNAs are NAD-capped, questioning the importance of NAD capping in yeast. This manuscript aims to map NAD-capped RNA in *Saccharomyces cerevisiae* with their original method. They detected >1400 species of NAD-RNAs, most of which are short less than 170 nt and low in frequency (>5%). RNA polymerase II accidentally uses NAD as a primer for transcription start. They identified a YAAG core motif in the promoter for efficient incorporation of NAD in the nascent chain. NAD-capped RNA is not capable for translation in vitro,

indicating that NAD-cap seems to be disadvantageous to the cell. The experiments were well executed, but data presentation and description should be improved for the revised version.

We thank the reviewer for his/her comments on the execution of our experiments and gladly follow the suggestions to improve data presentation and description.

Regarding RNA length and NAD capping, why was >170 nt chosen?

170 nt was the consequence of an arbitrary choice for our size selection of the cDNA amplicons by PAGE. By comparison with a DNA size standard, we selected amplicons between 150 and 300 bp, or between 300 and 500 bp. Since each amplicon contains the the Illumina sequencing primers (58+64=122 bp), and the 3'-end adapter together with 5'-end leading G (7+3 =10 bp), this leaves ~170 nt for the insert.

To clarify this issue, we changed a sentence in the first paragraph of the results part (p3, l. 62): "After adapter ligation, reverse transcription and PCR amplification, amplicons with sizes between 150 and 300 bp were selected by gel electrophoresis, thus this library represented mostly RNA species with sizes between 20 and 170 nt present in the original sample."

The authors need to show and compare size distribution of transcripts with and without NAD cap. If NAD capping occurs in short RNA, Northwestern blotting with Streptavidin HRP of the biotinylated total RNA will provide direct evidence for that.

We agree with the reviewer that the size distribution of transcripts with and without NAD cap must be carefully analyzed. We thoroughly considered the proposed Northwestern blotting experiment. Unfortunately, we conclude that the method does not have the required sensitivity for such a rare modification attached sparsely to thousands of cellular RNAs. For NAD captureSeq, we typically use 100 µg total RNA. For yeast samples, we obtain after ADPRC treatment, click biotinylation, reverse transcription and 17 cycles of PCR amplification ~25 ng ds cDNA. These 25 ng amplified cDNA correspond to ~190 fg RNA or 6 amol (6×10^{-18} mol, assuming an average length of 100 nt). After electrophoretic separation, these 6 amol will be distributed all over the lane (as they have different sizes) and not wind up in one sharp band. Assuming that almost all NAD-RNAs have sizes between 10 and 170 nt, ~40 zmol (4×10^{-20} mol) could be found per single-nt resolution band. This is several orders of magnitude below the detection limit of even the most sensitive chemifluorescent HRP assays (<https://doi.org/10.1016/j.jpha.2012.01.004>).

However, we have analyzed the size distributions of the cDNA used for sequencing (+ADPRC treated Sample group and -ADPRC Negative Control group) by Bioanalyzer analysis. This analysis revealed that smaller RNAs were significantly more abundant in the sample group, compared to the control. This finding indicates a smaller average size of NAD-RNAs, compared to non-NAD-RNAs. We added the following statement on p. 5 l. 96: "The preferential enrichment of smaller RNAs in NAD captureSeq was also confirmed by Bioanalyzer size analysis of the DNA amplicons (Supplementary Fig. 1j)." and added a panel to Supplementary Figure 1.

We would further like to note that our comparison of the percentage of full-length sequencing reads in the NAD captureSeq datasets (Supplementary Figure 6a) shows that the RNA species enriched in NAD captureSeq are on average shorter in the +ADPRC sample than in the -ADPRC control, again confirming the shorter length of NAD-RNAs.

In the abstract, it says >5% frequency of NAD capping. But, this reviewer cannot find any information on this figure in the manuscript.

The reviewer is right. We only mentioned the 5% value in the abstract and in the discussion, but not in the results part. We changed a sentence on p.7, l.147: "After normalization of the cp values to the same amount of input RNA in WT and *npv1Δ* and background subtraction, NAD modification ratios below 5%, in most cases between 1 and 3% were determined (Fig. 4c)."

They performed the NAD capture with total RNA isolated from WT and several KO strains. Quality of total RNA might be different in such mutant cell lines. They need to show pictures of electrophoresis or bioanalyzer data of each total RNA to confirm its quality.

We fully agree with the reviewer that the quality of the input RNA influences the outcome of sequencing experiments. We now provide electrophoresis and bioanalyzer data of total RNA for each replicate in each strain in the attached source data file (and for the reviewer's convenience copied at the end of this document). If Editor and/or reviewer think that they should be presented as a Supplementary Figure, we'll be happy to do so. We added a statement to the methods section (p. 17 l. 433): "The pellet was dissolved in 100 μ L dH₂O, the RNA concentration determined by Nanodrop and its integrity analyzed by both denaturing agarose gel electrophoresis and Bioanalyzer RNA integrity analysis."

Regarding Npy1 decapping, NAD capping was upregulated in only two transcripts (Fig. 2h), although NAD amount of total RNA decreased (Fig. 2f). The authors need to explain for this discrepancy.

We think that this discrepancy nicely illustrates the finding that NAD decapping does not happen randomly, and some mRNA 5'-ends are more susceptible to NPY1 decapping than others. As suggested by this reviewer, the two transcripts LSM7 and RSM10 seem to be particularly good substrates for NPY1 (see below), and may have been severely depleted during total RNA preparation in samples containing Npy1.

We added two sentences (see below).

LC/MS data for NAD cap should be shown in supplementary figure.

The standards and sample measurements for NAD cap measurement in LC/MS has now been supplemented in Supplementary Fig. 1i and Supplementary Fig. 2k.

mRNAs of LSM7 and RSM10 seem to be good substrates for Npy1 in the cell. The authors should analyze decapping efficiencies of these transcripts by recombinant Npy1 *in vitro*, and compare the activity with TDH3 mRNA.

Following the reviewer's suggestion, we have now included the Npy1 *in vitro* decapping kinetics on TDH3, LSM7, and RSM10 RNA in Fig. 4d,e. LSM7 and RSM10 are indeed better substrates, compared to TDH3. We added the following two sentences (p.7, l.152): "To test whether this increased modification ratio of the latter two transcripts might indicate that they are particularly good substrates for Npy1, being severely depleted in RNA preparation from cells containing this enzyme, we analyzed their decapping *in vitro*. Indeed, LSM7 and RSM10 were hydrolyzed much faster than TDH3 (Fig. 4d-e)."

Other comments

Poor information in figure legends.

For example, no information for bar graphs in Fig. 1ac.

The following sentence was added to the caption: "The bar chart represents the number of red and blue dots using the same color."

Pie chart in Fig. 1b. Is this a fraction of gene number or read number?

The reviewer is right. The following sentence was added to the caption: "The pie chart represents the number of reads assigned to each RNA class as percentage of total reads."

Additionally, we made the following changes to other Figure captions: Fig. 1d: "The chromosome number and genome locus are denoted at the top. "Fully treated" represents the library with ADPRC treatment, while "-ADPRC" stands for the library without this treatment." In Fig 2a: "Green bar heights represent relative transcript numbers in the sample group (+ADPRC) while grey bar heights represent the transcripts number from the same species in the negative control group (-ADPRC)."

Reviewer #3 (Remarks to the Author):

Review for the manuscript NCOMMS-20-22502-T by Dr. Jäschke and co-authors entitled "Extensive 5'-Surveillance Guards Against Non-Canonical NAD-Caps of Nuclear mRNAs in Yeast."

The ubiquitous redox coenzyme nicotinamide adenine dinucleotide (NAD) acts as a non-canonical cap structure on prokaryotic and eukaryotic ribonucleic acids. NAD-RNAs were found in human cells, and the NAD cap promotes RNA decay through DXO-mediated removal of 5' NAD. In yeast, only 3' RNA species and low abundance were reported. In this study, the whole landscape of NAD transcripts in yeast was demonstrated using the original NAD capture-Seq protocol. NAD is incorporated with low efficiency (<5%) during the initiation step by RNA polymerase II, which uses distinct promoters with a YAAG core motif. NAD-RNAs are mostly correspond to mRNA 5'-ends and NAD mRNAs are 3'-truncated and poorly translated. The authors discovered the Nudix pyrophosphohydrolase 1 (Npy1) as a new member of guard for NAD-RNA in yeast and performed a comprehensive analysis of three NAD-RNA decapping factors, Rai1, Dxo1, and Npy1. Three decapping enzymes guard against NAD-RNA at different cellular locations, targeting overlapping transcript populations. The authors propose that NAD incorporation into RNA seems to be disadvantageous to the cell, and different surveillance machinery is responsible for decapping and rejection of NAD-RNAs.

We thank the reviewer for accurately summarizing our main achievements.

The proposal is potentially interesting. However, the physiological relevance of the surveillance machinery for NAD-RNA is unclear. There is no direct evidence demonstrating that NAD incorporation into RNA seems to be disadvantageous to the cell. It is also unclear whether the defects in the clearance of NAD-RNAs directly cause RNA changes in the mutants. The growth defects RAI1 deletion may be due to the defects in removing the entire cap structure dinucleotide from an mRNA. Also, the deletion of the other two factors did not confer the significant defects in the growth. Therefore, there is no significant suggestion for the function of the NAD cap in yeast.

1) It was reported that Rai1 functions to clear mRNAs in cells subjected to glucose starvation or amino acid starvation by preferential hydrolysis of unmethylated capped RNA. To clarify the in vivo function of the scavenger system for the removal of NAD at the 5' end of mRNA, the authors should clarify the defects of the NAD-RNA's surveillance machinery in translation or RNA homeostasis. Since it is demonstrated that NAD mRNAs are poorly translated, it should be examined whether NAD-RNAs (TDH3, etc.) are rapidly degraded in cells subjected to stress conditions, including glucose starvation or amino acid starvation.

The interdependence of global cellular metabolism and the newly discovered class of RNA 5'-metabolite caps is of great interest to our lab and we appreciate the question. The opportunities for crosstalk between the metabolic state of the cell and gene expression could be manifold and the implications vast. Regulatory processes could be parsimoniously integrated, leveraging metabolic cofactors, covalently linked to the 5' end of respective transcripts. The exploration of this intriguing theme will therefore guide ongoing and prospective projects in our lab and those of our collaborators, aiming to enrich and partly redefine the nascent field of epitranscriptomics within the near future. We appreciate the Editor's judgement that such an analysis would exceed the scope of the current manuscript.

2) Given that mitochondria mRNAs were highly 5'-NAD modified, the authors propose a regulatory role of the NAD cap in mitochondria. Please discuss the possible function of three factors in the decapping of mitochondrial NAD-RNA.

The identity of the regulatory machinery dedicated to the 5' NAD cap of yeast mitochondrial RNAs, is indeed of utmost interest, given the prevalence of the modification in this organelle (Bird et al. eLife, 2018). Regarding the three nuclear-encoded decapping factors, Rai1, Dxo1 and Npy1 (the focus of this investigation), however, one would not expect their presence within the yeast mitochondrial matrix. This is because they lack a mitochondrial targeting sequence (MTS, an N-terminal amphipathic α -helix), required for the co-translational translocation of nascent polypeptides into the mitochondrial matrix, with MTS prediction scores of 0.031, 0.025 and 0.000 for Rai1, Dxo1 and Npy1, respectively. These scores were obtained using the MitoFates prediction algorithm (Fukasawa et al. Molecular & Cellular Proteomics, 2015), which ranges from 0 (absence of MTS predicted) to 1 (presence of MTS predicted). Furthermore, empirical proteomic data indicate that the three decapping enzymes could not be found among the set of yeast mitochondrial proteins (Morgenstern

et al. Cell Reports, 2017, Table S3) – in contrast to the *S. cerevisiae* Nudix hydrolase Ysa1, an ADP-ribose pyrophosphatase that also catalyzes the hydrolysis of NADH *in vitro* (Dunn et al. JBC, 1999), and reaches an MTS prediction score of 0.996. Taken together, direct contact of NAD-modified RNAs, transcribed from the mitochondrial genome, and the three decapping enzymes investigated in this study seems unlikely.

A sentence capturing these considerations has been appended on page 14, l. 348:

“Also, the identity of a dedicated surveillance machinery within the mitochondrial matrix is of considerable interest. The three decapping enzymes investigated here, however, are unlikely to encounter mitochondrial transcripts, as they lack the required targeting sequence and are absent in yeast mitochondrial proteomic data (Fukasawa et al. Molecular & Cellular Proteomics, 2015; Morgenstern et al. Cell Reports, 2017).”

3) It was reported that DXO/Rai1 enzymes remove 5'-end FAD and dephospho-CoA caps on RNAs. Is it possible to discuss the possibility that Npy1 to remove the modifications at 5'-end of RNAs?

Thank you for this intriguing question. Indeed, *S. cerevisiae* Npy1 was previously reported to exhibit residual activity in hydrolyzing the free redox cofactor flavin adenine dinucleotide (FAD), with a relative enzymatic activity of 9.5% in comparison to its main substrate NADH (Abdelraheim et al. Archives of Biochemistry and Biophysics, 2001). This initial hint renders the question worthy for future investigation – which should closely follow the example set by two pioneering works, published earlier this year from the labs of Mike Kiledjian and Liang Tong (Doamekpor et al. NAR, 2020 and Sharma et al. NAR, 2020). The manuscript has been correspondingly altered, with the addition of the two sentences (page 16, l. 394):

“An earlier report, providing evidence that dinucleotide hydrolysis mediated by *S. cerevisiae* Npy1 is not entirely restricted to NAD, but also includes the redox cofactor flavin adenine dinucleotide (FAD) among others (Abdelraheim et al. Archives of Biochemistry and Biophysics, 2001), warrants an in-depth biochemical characterization to define the set of RNA 5' metabolite caps, targeted by this enzyme. A corresponding study should hereby follow the example set by the systematic and meticulous elucidation of RNA 5' cap specificities of mammalian DXO and *Schizosaccharomyces pombe* Rai1 (Doamekpor et al. NAR, 2020 and Sharma et al. NAR, 2020).”

4) Please discuss the putative mechanism for RNAP II to select a different TSS for initiating transcription with NAD, and possible mechanism to recruit Rai1 to the RNAP elongation complex that initiates from TSS.

Thank you for this thought-provoking suggestion, we hope the following paragraph (added on page 16, l. 387) might embed our findings more thoroughly within the interesting field of 5'-end cap quality control, which is highly relevant to gain a deeper understanding concerning the results of our efforts:

‘The Rat1-Rai1 complex is hereby believed to play an important role, mediating 5'-end cap quality control (5'QC) in the yeast RNAP II transcription cycle, following the transcription checkpoint pause stage, whereby RNAP II enters transcription elongation upon phosphorylation of distinct serines within the C-terminal repeat domain of the polymerase (Kiledjian et al. The Enzymes, 2012). Surveillance and hydrolysis of the accidentally incorporated 5'-NAD cap could be enacted in a mechanistically similar manner, mirroring the clearance of unmethylated, aberrantly capped mRNAs by the Rat1-Rai1 heterodimer (Jiao et al. Nature, 2010 and Xiang et al. Nature, 2009).’

In addition, the following two sentences were added (page 15, l. 370):

“An increased affinity of RNAP II for the YAAG motif, while the polymerase transiently harbors NAD in its catalytic site, could potentially also explain the enrichment of non-canonical TSSs upon execution

of NAD captureSeq. An in-depth biochemical investigation should explore the possibility of NCIN-mediated guidance of RNAP II, and other RNA polymerases, to distinct TSSs.”

Source Figure for reviewers.

a-d. Total RNA in denaturing agarose gel electrophoresis. The replicate “x” was denoted as “_x”. 100 bp DNA marker is shown in the left lane. DNase I treatment was conducted as described in the methods section. The gels were stained by ethidium bromide. Biological triplicates.

e. Bioanalyzer electrophoresis of total RNA from a-d.

REVIEWERS' COMMENTS:

Reviewer #1 (Remarks to the Author):

The authors have adequately addressed my concerns from the prior round of review. The manuscript is suitable for publication.

Reviewer #2 (Remarks to the Author):

The authors adequately addressed to my concerns in this revised version. I have no further comment.

Reviewer #3 (Remarks to the Author):

Review for the manuscript NCOMMS-20-22502A by Dr. Jäschke and co-authors entitled "Extensive 5'-Surveillance Guards Against Non-Canonical NAD-Caps of Nuclear mRNAs in Yeast."

The authors have addressed most of my previous concerns. I support the publication of manuscript in Nature Communications.

Manuscript NCOMMS-20-22502-A

Point-by-point response to the reviewers' comments

Reviewer #1 (Remarks to the Author):

The authors have adequately addressed my concerns from the prior round of review. The manuscript is suitable for publication.

Reviewer #2 (Remarks to the Author):

The authors adequately addressed to my concerns in this revised version. I have no further comment.

Reviewer #3 (Remarks to the Author):

Review for the manuscript NCOMMS-20-22502A by Dr. Jäschke and co-authors entitled "Extensive 5'-Surveillance Guards Against Non-Canonical NAD-Caps of Nuclear mRNAs in Yeast."

The authors have addressed most of my previous concerns. I support the publication of manuscript in Nature Communications.

We thank all three reviewers for the positive assessment and the recommendation to publish the work in Nature Communications.